# Your contrastive learning problem is secretly a distribution alignment problem

**Zihao Chen**[*], **Chi-Heng Lin, Ran Liu, Jingyun Xiao, Eva L. Dyer**[*]
School of Electrical & Computer Engineering
Georgia Tech, Atlanta, GA

## Abstract

Despite the success of contrastive learning (CL) in vision and language, its theoretical foundations and mechanisms for building representations remain poorly understood. In this work, we build connections between noise contrastive estimation losses widely used in CL and distribution alignment with entropic optimal transport (OT). This connection allows us to develop a family of different losses and multistep iterative variants for existing CL methods. Intuitively, by using more information from the distribution of latents, our approach allows a more distribution-aware manipulation of the relationships within augmented sample sets. We provide theoretical insights and experimental evidence demonstrating the benefits of our approach for *generalized contrastive alignment*. Through this framework, it is possible to leverage tools in OT to build unbalanced losses to handle noisy views and customize the representation space by changing the constraints on alignment. By reframing contrastive learning as an alignment problem and leveraging existing optimization tools for OT, our work provides new insights and connections between different self-supervised learning models in addition to new tools that can be more easily adapted to incorporate domain knowledge into learning.

## 1 Introduction

In machine learning, the availability of vast amounts of unlabeled data has created an opportunity to learn meaningful representations without relying on costly labeled datasets [26, 52, 27]. Self-supervised learning has emerged as a powerful solution to this problem, allowing models to leverage the inherent structure in data to build useful representations. Among self-supervised methods, contrastive learning (CL) is widely adopted for its ability to create robust representations by distinguishing between similar (positive) and dissimilar (negative) data pairs. With success in fields like image and language processing [8, 46], contrastive learning now also shows promise in domains where cross-modal, noisy, or structurally complex data make labeling especially challenging [34, 56, 10].

Traditional contrastive learning methods primarily aim to bring positive pairs—often augmentations of the same sample—closer together in representation space. While effective, this approach often struggles with real-world challenges such as noise in views, variations in data quality, or shifts introduced by complex transformations, where positive pairs may not perfectly align. Additionally, in tasks requiring domain generalization, aligning representations across diverse domains (e.g., variations in style or sensor type) is critical but difficult to achieve with standard contrastive learning, which typically lacks mechanisms for incorporating domain-specific relationships. These limitations highlight the need for a more flexible approach that can adapt alignment strategies based on the data structure, allowing for finer control over similarity and dissimilarity among samples.

---

[*]Contact: {zchen959, evadyer}@gatech.edu

38th Conference on Neural Information Processing Systems (NeurIPS 2024).

To address this challenge, we introduce a novel *generalized contrastive alignment* (GCA) framework, which reinterprets contrastive learning as a distributional alignment problem. Our method allows flexible control over the alignment of samples by defining a target transport plan, $\mathbf{P}_{tgt}$, that serves as a customizable alignment guide. For example, setting $\mathbf{P}_{tgt}$ to resemble a diagonal matrix encourages each positive to align primarily with itself or its augmentations, thereby reducing the effect of noise between views. Alternatively, we can incorporate more complex constraints, such as weighting alignments based on view quality or enforcing partial alignment structures where noise or data heterogeneity is prevalent. This flexibility enables GCA to adapt effectively to a wide range of tasks, from simple twin view alignments to scenarios with noisy or variably aligned data.

Our approach also bridges connections between GCA and established methods, such as InfoNCE (INCE) [38], Robust InfoNCE (RINCE) [12], and BYOL [22], demonstrating that these can be viewed as iterative alignment objectives with Bregman projections [6, 21]. This perspective allows us to systematically analyze and improve uniformity within the latent space, a property that enhances representation quality and ultimately boosts downstream classification performance.

We validate our method through extensive experiments on both image classification and noisy data tasks, demonstrating that GCA's unbalanced OT (UOT) formulations improve classification performance by relaxing our constraints on alignment. Our results show that GCA offers a robust and versatile framework for contrastive learning, providing flexibility and performance gains over existing methods and presenting a promising approach to addressing different sources of variability in self-supervised learning.

The contributions of this work include:

- A new framework called *generalized contrastive alignment* (GCA), which reinterprets standard contrastive learning as a distributional alignment problem, using optimal transport to provide flexible control over alignment objectives. This approach allows us to derive a novel class of contrastive losses and algorithms that adapt effectively to varied data structures and build customizable transport plans.

- We present GCA-UOT, a contrastive learning method that achieves strong performance on standard augmentation regimes and excels in scenarios with more extreme augmentations or data corrupted by transformations. GCA-UOT leverages unbalanced transport to adaptively weight positive alignments, enhancing robustness against view noise and cross-domain variations.

- We provide theoretical guarantees for the convergence of our GCA-based methods and show that our alignment objectives improve representation quality by enhancing the uniformity of negatives and strengthening alignment within positive pairs. This leads to more discriminative and resilient representations, even in challenging data conditions.

- Empirically, we demonstrate the effectiveness of GCA in both image classification and domain generalization tasks. Through flexible, unbalanced OT-based losses, GCA achieves superior classification performance and adapts alignment to include domain-specific information where relevant, without compromising classification accuracy in domain generalization.

## 2  Background

### 2.1  Contrastive learning

Contrastive learning (CL) is a representation learning methodology that uses positive and negative pairs to define similarity in the latent space. Let $\mathcal{D} = \{\mathbf{x}_i\}_{i=1}^N$ denote our dataset. For each sample $\mathbf{x}_i$ in a batch of training data with size $B$, we create two augmented copies $\mathbf{x}_i'$ and $\mathbf{x}_i''$ independently, i.e., $\mathbf{x}_i' = \psi(\mathbf{x}_i)$ where $\psi$ is a randomly drawn augmentation function from some augmentation class $\mathcal{A}$ and likewise for $\mathbf{x}_i''$. The $(\mathbf{x}_i', \mathbf{x}_i'')$ is called a positive pair of $\mathbf{x}_i$ while $(\mathbf{x}_i', \mathbf{x}_j'')$ is treated as a negative pair for any $j \neq i$. One of the most widely used formulations of the CL problem, InfoNCE (INCE) [8], seeks to maximize the negative log probability that a sample is correctly classified as

$$\mathcal{L}_{\text{INCE}} = -\log\left(\frac{e^{s_{ii}}}{e^{s_{ii}} + \sum_{i \neq j} e^{s_{ij}}}\right), \tag{1}$$

where $s_{ij} = \varepsilon^{-1} f_\theta(\mathbf{x}_i')^\top f_\theta(\mathbf{x}_j'') / \|f_\theta(\mathbf{x}_i')\| \|f_\theta(\mathbf{x}_j'')\|$ is the score between augmented samples.

Building upon the principles of INCE, SimCLR [8] and MoCo [24] are two representative works that form the foundation of contrastive learning methods for visual representation tasks. Alternatively, BYOL [22] and SimSiam [9] discard the use of negative samples to avoid large batch size and instead use exponential moving average-based updates to avoid representational collapse. Recent contrastive methods have focused on improving the tolerance to noise in samples to enhance robustness in diverse scenarios [13]. Among them, Robust INCE (RINCE) is a robust contrastive loss function characterized by its symmetric properties and theoretical resistance to noisy labels [47, 12]. Specifically, RINCE provides robustness to noisy views by introducing adjustable parameters $\lambda$ and $q$ [12] which rebalance the cost of positive and negative views, resulting in the following loss:

$$\mathcal{L}_{\text{RINCE}}^{\lambda,q} = \frac{1}{q}\left(-e^{qs_{ii}} + \lambda^q(e^{s_{ii}} + \sum_{i\neq j} e^{s_{ij}})^q\right) \tag{2}$$

By optimizing the above loss functions, the encoder $f$ is trained to construct a semantically coherent representation space where positive pairs of samples are positioned nearby, while those negative pairs with divergent semantic attributes are separated [57].

## 2.2 Proximal Operators and Projections

To make the connections between different CL losses clearer later, we use the notion of proximal operators. In words, the proximal operator will provide a way to find the closest point in some closed convex set. Formally, we can define the proximal operator as follows.

**Definition 1** (Proximal Operator). *Let $d_\Gamma(\mathbf{x}, \mathbf{v}) = \Gamma(\mathbf{x}) - \Gamma(\mathbf{v}) - \langle \nabla\Gamma(\mathbf{v}), \mathbf{x} - \mathbf{v}\rangle$ be a Bregman divergence with a convex function $\Gamma$. The proximal operator of $h : \mathcal{X} \to \mathbb{R} \cup \{+\infty\}$ is defined for a point $\mathbf{v} \in \mathcal{X}$ with a closed convex set $\mathcal{B} \subseteq \mathcal{X}$ :*

$$Prox_{h,\mathcal{B}}^{d_\Gamma}(\mathbf{v}) = \arg\min_{\mathbf{x}\in\mathcal{B}} \{h(\mathbf{x}) + d_\Gamma(\mathbf{x}, \mathbf{v})\}.$$

Moreover, we can define the concept of a projection as a special case of the proximal operator when we let $h(\mathbf{x})$ be an indicator function $h_{\mathcal{B}}(x) = \{0, \text{if } x \in \mathcal{B}; \infty, \text{if } x \notin \mathcal{B}\}$ on constraint set $\mathcal{B}$. See Appendix A.2 for more details.

## 2.3 Solving Optimal Transport Through Proximal Point Methods

Optimal transport (OT) is widely used in characterizing the distance between two collections of samples $\{\mathbf{x}_i\}_{i=1}^B$ and $\{\mathbf{y}_j\}_{j=1}^B$ with associated measures $\mu = \sum_{i=1}^B \delta_{\mathbf{x}_i} p_i$ and $\nu = \sum_{j=1}^B \delta_{\mathbf{y}_j} q_j$ with Dirac delta function $\delta_{\mathbf{x}}$ and $\delta_{\mathbf{y}}$ on finite support [43]. Here, $p$ and $q$ are vertices of the $\mathbb{R}^B$ simplex defined as $\Delta_B := \{v \in \mathbb{R}^B : v_i \geq 0, \sum_{i=1}^B v_i = 1\}$. OT aims to learn a joint coupling matrix, or transport plan $\mathbf{P} \in \mathbb{R}_+^{B\times B}$ that minimizes the cost of transporting mass encoded by cost matrix $\mathbf{C} \in \mathbb{R}_+^{B\times B}$, from one distribution to another. In practice, entropy regularization is used to solve the OT objective, resulting in the following entropy-regularized OT (EOT) objective:

$$\min_{\mathbf{P}\in\mathcal{B}} \langle\mathbf{P}, \mathbf{C}\rangle - \varepsilon H(\mathbf{P}), \quad \text{where } H(\mathbf{P}) = -\sum_{ij}\mathbf{P}_{ij}\log(\mathbf{P}_{ij}), \tag{3}$$

where $\varepsilon$ is a user specified parameter that controls the amount of smoothing in the transport plan, and $\mathbf{C}(\mathbf{x}, \mathbf{y}) = 1 - \langle\mathbf{x}, \mathbf{y}\rangle/\|\mathbf{x}\|\|\mathbf{y}\|$ is often set to encode the cosine similarity between pairs of samples.

**The Sinkhorn Algorithm and its Interpretation as a Bregman Projection.** Solving Equation (3) could be interpreted as iterative alignment problem on a Hilbert space generated from the kernel $\mathbf{K}_{ij} = \exp(-\mathbf{C}_{i,j}/\varepsilon)$. This alignment problem can be solved through iterative Bregman projections onto the two constraints sets that encode the marginals along the rows and columns [3, 5, 43]:

$$C_1^\mu := \{\mathbf{P} : \mathbf{P}\mathbb{1}_B = \mu\}, C_2^\nu := \{\mathbf{P} : \mathbf{P}^\top\mathbb{1}_B = \nu\} \tag{4}$$

The first step of Bregman projection is to find the minimizer $\mathbf{P}^{(1)} = \arg\min\{\varepsilon\text{KL}(\mathbf{P}\|\mathbf{K}) : \mathbf{P}\mathbb{1}_B = \mu\}$ by the proximal operator $Prox_{C_1^\mu}^{\text{KL}}(\mathbf{K})$ with Lagrange multiplier $f$ on the row constraint set $\mathcal{C}_1^\mu$, and compute its derivatives with respect to $\mathbf{P}$ with $\mathbf{u} = e^{f/\varepsilon} > 0$:

$$\varepsilon\log(\mathbf{P}^{(1)}/\mathbf{K}) - f\mathbb{1} = 0 \Rightarrow \mathbf{P}^{(1)} = \mathbf{u}\mathbf{K}, \quad \langle\mathbf{P}^{(1)}, \mathbb{1}\rangle = \mu \Rightarrow \langle\mathbf{u}\mathbf{K}, \mathbb{1}\rangle = \mu, \mathbf{u} = \frac{\mu}{\mathbf{K}\mathbb{1}} \tag{5}$$

Next, we project $\mathbf{P}^{(1)}$ onto the column constraint set $C_2^\nu$, resulting in $\mathbf{P}^{(2)} \coloneqq \mathrm{Prox}_{C_2^\nu}^{\mathrm{KL}}(\mathbf{P}^{(1)}) = \mathbf{P}^{(1)}\mathrm{diag}(\frac{\nu}{\mathbf{P}^{(1)\top}\mathbb{1}_B})$. The iterative updates can be succinctly expressed as the Sinkhorn iterations:

$$\mathbf{P}^{(2t+1)} = \mathrm{diag}(\mathbf{u}^{(t+1)})\mathbf{K}\,\mathrm{diag}(\mathbf{v}^{(t)}), \quad \mathbf{P}^{(2t+2)} = \mathrm{diag}(\mathbf{u}^{(t+1)})\mathbf{K}\,\mathrm{diag}(\mathbf{v}^{(t+1)}), \qquad (6)$$

with the scaling vectors $\mathbf{u}^{(t)}$ and $\mathbf{v}^{(t)}$ updated according to:

$$\mathbf{u}^{(t+1)} \overset{\mathrm{def}}{=} \frac{\mu}{\mathbf{K}\mathbf{v}^{(t)}}, \quad \mathbf{v}^{(t+1)} \overset{\mathrm{def}}{=} \frac{\nu}{\mathbf{K}^{\mathrm{T}}\mathbf{u}^{(t)}}. \qquad (7)$$

Here, iterations converge to a stable transport plan $\mathbf{P}^{(\infty)}$ as the optimal solution of Equation (3), which provides the minimum cost matching between two distributions. The convergence and dynamics of OT and its dual formulation have been studied extensively in [4, 43, 19, 1]. Thus, these results guarantee that the iterates will converge to the optimal solution of the EOT objective, or that $\mathbf{P}^{(t)} \to \mathbf{P}^{(\infty)}$ with $t \to \infty$. See Appendix A.3 for more details on both the continuous and discrete formulations of OT.

## 2.4 Wasserstein Dependency Measure

The Wasserstein Dependency Measure (WDM) is a measure of deviation between two probability measures. We will use this later and thus provide the formal definition here [39].

**Definition 2** (Wasserstein Dependency Measure). *Define the WDM as the Wasserstein distance ($W_1$) between the joint distribution $\pi(x,y)$ and the product of marginal distributions $\mu \otimes \nu(x,y)$ of two random variables $x$ and $y$. $W_1(\pi, \mu \otimes \nu) = \sup_{f \in \mathcal{C}(\mathcal{X} \times \mathcal{Y})} \left( \mathbb{E}_{\pi(x,y)}[f(x,y)] - \mathbb{E}_{\mu \otimes \nu(x,y)}[f(x,y)] \right)$, where $\mathcal{C}(\mathcal{X} \times \mathcal{Y})$ denotes the set of all 1-Lipschitz functions from $\mathcal{X} \times \mathcal{Y}$ to $\mathbb{R}$.*

## 2.5 Optimal Transport and Alignment in Representation Learning

Distribution alignment and OT have been widely used for domain adaptation [33, 14, 30, 59], and in generative modeling [2, 55, 49, 58]. The connections between distribution alignment and contrastive learning, however, are still nascent. In [51], the authors explore the connection between inverse OT (IOT) [32, 53, 18] and INCE. Our work builds on this connection to OT to build robust divergences (RINCE) and to build a novel unbalanced optimal transport (UOT) method (Section 3.3). Additionally, we show how our framework can be used to build flexible methods for encouraging contrast at multiple levels. We use this concept of hierarchical contrast and show that it can be used in domain generalization settings (Section 6.2). It is of note that GCA-UOT focuses on relaxing the hard constraints on the row and columns into the soft penalties, which is different with the idea of "unbalanced matching" in [51] which considers the case where the encoders may not have the same weights.

# 3 Generalized Contrastive Alignment (GCA)

In this section, we will introduce a new framework for *generalized contrastive alignment* and demonstrate the connections between contrastive learning and optimal transport.

## 3.1 Problem Formulation

Traditional contrastive learning methods focus on bringing positive examples, such as augmentations of the same sample, closer together in representation space. In contrast, our approach reframes contrastive learning as a distributional alignment problem, allowing flexible control over how pairs are matched by imposing specific constraints on the target transport plan, $\mathbf{P}_{tgt}$.

Our objective is to learn an encoder $f_\theta$ that minimizes the *transport cost* between positive samples. By defining $\mathbf{P}_{tgt}$ with specific alignment rules, such as domain-specific or hierarchical constraints, we can influence how samples are organized in the latent space. For instance, setting $\mathbf{P}_{tgt}$ to resemble a diagonal matrix encourages each positive to align primarily with itself or its augmentations, minimizing $\mathrm{div}(\mathbf{I}||\mathbf{P}) \approx 0$, where div measures the deviation from an identity matrix (e.g., KL-divergence).

This flexibility allows us to encode more nuanced forms of similarity, adapting to tasks where alignment structure varies based on domain, class, or other high-level constraints. By expanding contrastive learning in this way, our method enhances separation of negatives while addressing complex relational patterns, making it suitable for a wider range of learning tasks.

**Defining the Kernel Space.** Before formally stating our objective, we first need to define the concept of an augmentation kernel for our positive and negative examples.

**Definition 3** (Augmentation Kernel)**.** *Let $f_\theta$ denote an encoder with parameters $\theta$ and let $(\mathbf{x}'_i, \mathbf{x}''_j) \sim \mathcal{A}$ be two views drawn from the family of augmentations $\mathcal{A}$. The augmentation kernel for the encoder $\theta$ is defined as $\mathbf{K}_\theta(\mathbf{x}'_i, \mathbf{x}''_j) = \exp(-dist(\widetilde{f}_\theta(\mathbf{x}'_i), \widetilde{f}_\theta(\mathbf{x}''_j))/\varepsilon)$, where $dist(\cdot)$ can be an arbitrary distance metric, and $\widetilde{f}_\theta(\mathbf{x}'_i)$ is the normalized output of $f_\theta$, and $\varepsilon$ is the regularization parameter.*

**Main Objective.** With this definition in hand, we can now formalize our objective as follows:

$$\min_\theta \; d_M\big(\mathbf{P}_{\text{tgt}}||\mathbf{P}_\theta\big), \;\; \text{with} \quad \mathbf{P}_\theta = \arg\min_{\mathbf{P}\in\mathcal{B}}\{h(\mathbf{P}) + d_\Gamma(\mathbf{P}||\mathbf{K}_\theta)\}, \tag{8}$$

where $\mathbf{K}_\theta$ is the augmentation kernel defined in Definition (3), $h(x)$ is a convex function (typically an indicator function), $\mathcal{B}$ is a closed convex constraint set (i.e. Birkhoff polytope) that defines the constraints of proximal operators, $d_\Gamma$ is a Bregman divergence that is used to find the nearest points $\mathbf{P}_\theta$ on the constraint set $\mathcal{B}$ of $\mathbf{K}_\theta$, $d_M$ is a convex function (e.g., KL-divergence) that measures divergence between $\mathbf{P}_\theta$ and the target coupling plan $\mathbf{P}_{\text{tgt}}$.

Our objective is a bi-level optimization problem which aims to learn a representation that minimizes the divergence between the transport plan $\mathbf{P}_\theta$ with the target alignment plan $\mathbf{P}_{\text{tgt}}$ that encodes the matching constraints. When we consider a standard contrastive learning setup where we have pairs of positive examples the source and target distribution, then the target $\mathbf{P}_{\text{tgt}}$ is the identity matrix $\mathbf{I}$. However, we will show later that other alignment constraints can be considered. Moreover, when $\mathcal{B}$ is the intersection of more constraint sets like $C_1^\mu \cap C_2^\nu$ in Equation (4), a nature way to get the approximation of the nearest points $\mathbf{P}_\theta$ of $\mathbf{K}_\theta$ is to run iterative projections algorithm [3], which could be extended into the intersection of several constraint sets like $\{\cap_{i=1}^n C_i\}$, resulting in a multi-marginal problem [41].

### 3.2 A Proximal Point Algorithm for GCA

In practice, we can solve the alignment problem above by iteratively updating the two main components in our bi-level objective. First, for a fixed encoder parameters $\theta$, we obtain the transport coupling $\mathbf{P}_\theta$ through our corresponding proximal operator. Second, we measure the deviation between the transport plan $\mathbf{P}_\theta$ with the target $\mathbf{P}_{\text{tgt}}$ that encodes our matching constraints, which denotes the ideal alignment plan on the intersection of the constraint sets. We provide pseudocode for this iterative approach in Algorithm 1, which we refer to as generalized contrastive alignment or GCA. The implementation of our methods is in https://github.com/nerdslab/gca.

---

**Algorithm 1** Proximal-Point Algorithm for Generalized Contrastive Alignment (GCA)

---

1: **Initialization:** Initial encoder parameters $\theta$, target transport plan $\mathbf{P}_{\text{tgt}}$, kernel function $\mathbf{K}_\theta$, the function $h(x)$, divergences $d_\Gamma$ and $d_M$ (KL or $W_1$). Initialize transport plan $\mathbf{P}_\theta$ based on $\theta$.
2: **Compute the transport coupling $\mathbf{P}_\theta$:** Update $\mathbf{P}_\theta$ using the proximal operator scaling for fixed $\theta$ as described in Eq. (8):

$$\mathbf{P}_\theta = \arg\min_{\mathbf{P}\in\mathcal{B}}\{h(\mathbf{P}) + d_\Gamma(\mathbf{P}||\mathbf{K}_\theta)\}.$$

3: **Calculate the loss:** Calculate deviation between the target and current transport plans

$$\mathcal{L}_{GCA} = d_M(\mathbf{P}_\theta, \mathbf{P}_{\text{tgt}}).$$

   Update networks $f_\theta$ (encoder) and $g_\theta$ (projector) to minimize $\mathcal{L}_{GCA}$.
4: **Repeat until convergence:** Repeat steps 2 and 3 until convergence.

---

Computing the transport coupling $\mathbf{P}_\theta$ [2] (forward-pass) in GCA algorithms could be treated as a specific type of Dykstra's projection algorithms [5], which computes the **iterative projection** on the intersection of affine convex sets [3, 42]. The proofs of convergence are provided in Appendix B.1.

---

[2]With a single constraint set like $C_1^\mu$ in Equation (4), computing the proximal point only involves a single projection. However, if there are intersecting constraint sets like $C_1^\mu \cap C_2^\nu$, solving for the proximal point requires multiple projections before we approach the nearest point on their intersection.

### 3.3 GCA-UOT Method

We can also benefit from the rich literature on optimal transport to build different relaxations of our objective [43, 7, 54, 33, 35]. In particular, we choose to leverage a formulation of *unbalanced optimal transport* (UOT) to further relax the marginal constraints [11] in our objective.

In this case, we can add the dual form of $d_\Gamma$ to the Equation (8) and reformulate our objective as:

$$\min_\theta \; d_M(\mathbf{P}_{\text{tgt}}\|\mathbf{P}_\theta) + \lambda_1 \mathbf{h}_\mathcal{F}(\mathbf{P}_\theta \mathbb{1}\|\mu) + \lambda_2 \mathbf{h}_\mathcal{G}(\mathbf{P}_\theta^\top \mathbb{1}\|\nu) + \varepsilon \mathbf{H}(\mathbf{P}_\theta). \tag{9}$$

Here $h_\mathcal{F}$ and $h_\mathcal{G}$ can be different divergence measures (e.g., KL divergence) that penalize deviations from the desired marginals $\mu$ and $\nu$, and $\lambda_1$ and $\lambda_2$ are regularization parameters that control the trade-off between the transport cost and the divergence penalties. This relaxation leads to different types of proximal operators which we outline in Appendix B.2. The impact of the entropy regularization parameter $\varepsilon$ on the coupling matrix is studied in Figure A5, along with the number of iterations and corresponding sensitivity is provided in Figure A6.

### 3.4 Modifying the Target Transport Plan to Encode Matching Constraints

Contrastive learning objectives can be cast as a minimization of the deviations between the transport plan $\mathbf{P}_\theta$ and the identity matrix, i.e., $\mathbf{P}_{tgt} = \mathbf{I}$. However, our GCA formulation enables learning representations that extend beyond this one-to-one matching constraint. This flexibility allows us to incorporate additional matching constraints informed by domain-specific knowledge. For example, in domain generalization scenarios [23, 28], where each batch contains samples from multiple domains, the target alignment plan can be structured as:

$$\mathbf{P}_{\text{tgt}}[i, j] = \mathbf{I}[i, j] + \alpha \cdot \mathbb{I}(D_i = D_j, \, i \neq j) + \beta \cdot \mathbb{I}(D_i \neq D_j, \, i \neq j),$$

Where $\mathbb{I}(\cdot)$ is the indicator function, which equals 1 if the condition inside is true and 0 otherwise. $D_i$ represents the domain of sample $i$, where $\alpha \geq 0$ and $\beta \geq 0$. In this case, we can improve the representation by building the block constraints which encode either class information (in supervised setting) or domain information (in across domain generalization, visualized in Figure 1).

### 3.5 Computational Complexity

The forward-pass only involves the scaling operations in Equation (7) and doesn't affect the complexity of the backward-pass. Therefore, GCA methods can be thought of as a form of batch normalization operations with adaptive scaling. An analysis of the complexity is provided along with experiments in Appendix B.1. Our results show that GCA iterations only slightly increase the computational complexity when compared with their single step equivalent (GCA-INCE vs. INCE). However, we found that GCA-UOT is faster than INCE due to the improved symmetry and smoothness of the loss. Moreover, we record the floating point operations per second (Flops) of running GCA methods. We find that GCA-INCE (6.65 MFlops) has $5\%$ more Flops than INCE (6.31 MFlops), while GCA-UOT saves $30\%$ Flops (4.54 MFlops). These results show that our GCA-UOT method is not only superior in terms of accuracy but also in speed.

## 4 Building Connections to Different CL Objectives

In this section, we show how the modification of the different parts of our main objective $(d_\Gamma, d_M, \mathcal{B}, \mathbf{K}_\theta)$ in Equation (8) can be connected to different contrastive losses. See Table 1 for a summary of how different losses can be mapped back to our formulation.

Table 1: *Comparison of different contrastive alignment objectives.* Here we have $C_1'^\mu$ and $C_2^\nu$ as constraint sets (denoted as $\mathcal{B}$) defined in Equation (4) with their corresponding indicator function. "Iter" refers to iterative methods.

| Methods | $d_M$ | $d_\Gamma$ | $\mathcal{B}$ | Iter |
|---|---|---|---|---|
| INCE | KL | KL | $C_1^\mu$ | |
| GCA-INCE | KL | KL | $C_1^\mu \cap C_2^\nu$ | ✓ |
| RINCE (q=1) | W1 | KL | $C_1^\mu$ | |
| GCA-RINCE (q=1) | W1 | KL | $C_1^\mu \cap C_2^\nu$ | ✓ |
| BYOL | KL | L2 | $R^{B \times B}$ | |

### 4.1 Connection to INCE

An interesting connection that we can make between GCA main objective and contrastive learning is that we can interpret INCE as a **single step** in a iterative GCA objective [51]. This connection can be further summarized through the following theorem.

**Theorem 1** (INCE Equivalence). *Let $\mathbf{K}_\theta$ denote the augmentation kernel as in Definition* (3) *with cosine similarity, $d_\Gamma$ and $d_M$ equal to KL-divergence, and constraint set as $C_1^\mu$ in Equation* (4). *The INCE objective in Equation* (1) *can be re-expressed as a GCA problem in Equation* (8) *as follows:*

$$\min_\theta KL\big(\mathbf{I}||Prox_{C_1^\mu}^{KL}(\mathbf{K}_\theta)\big). \tag{10}$$

The proof is contained in Appendix B.3. Theorem (1) shows that the INCE loss can be viewed as solving the matching problems in Equation (3) with row normalization constraints $C_1^\mu$. This connection between GCA and INCE allows us to derive the iterative algorithm for GCA-INCE by running Bregman projection iteratively on both row and column normalization sets

## 4.2 Connection to RINCE

We introduce the following result to build the connection between our framework and RINCE [12].

**Theorem 2** (RINCE Equivalence). *Let $\mathbf{K}_\theta$ denote the augmentation kernel as in Definition* (3). *Set target plan $\mathbf{P}_{tgt} = \mathbf{I}$, $d_\Gamma$ equal to the KL-divergence, $d_M(\mathbf{I}||\mathbf{P}) = -\frac{1}{q}(\frac{\text{diag}(\mathbf{P}_\theta)}{\mathbf{u}})^q + (\frac{\lambda \mathbf{I}}{\mathbf{u}})^q$ with $\lambda$, $q$, and $\mathbf{u} = \text{diag}\big(\frac{\mu}{\mathbf{P}^{(0)}\mathbb{1}}\big)$, and constraint set $C_1^\mu$ defined in Equation* (4). *The RINCE objective in Equation* (2) *can be re-expressed as a GCA problem as follows:*

$$\min_\theta d_M(\mathbf{I}||\mathbf{P}_\theta), \quad \text{with} \ \ \mathbf{P}_\theta = Prox_{C_1^\mu}^{KL}(\mathbf{K}_\theta), \tag{11}$$

The proof is provided in Appendix B.4.1. As we can see, RINCE introduces adjustable parameters $q$ and $\lambda$, with $\lambda$ controlling the weight of negative samples, while $q \in (0, 1]$ serves to switch between KL divergence and Wasserstein discrepancy. When $q = 1$, we have the following theorem:

**Theorem 3** (W1 Equivalence). *Let $\mathbf{K}_\theta$ denote the augmentation kernel as in Definition* (3) *with cosine similarity. Set target plan $\mathbf{P}_{tgt} = \mathbf{I}$, $d_\Gamma$ equal to the KL-divergence, $d_M$ equal to the 1-Wasserstein distance $(W_1)$ in Definition* (2), *and the constraint set as $C_1^\mu$ defined in Equation* (4). *The RINCE object in Equation* (2) *with $q = 1$ can be re-expressed as a GCA problem as follows:*

$$\min_\theta W_1\big(\mathbf{P}_{tgt}||Prox_{C_1^\mu}^{KL}(\mathbf{K}_\theta)\big). \tag{12}$$

See Appendix B.5 for the proof.

This connection to RINCE suggests an extended iterative formulation to calculate the coupling plan as the projection point $\mathbf{P}^{(\infty)} = Prox_{C_1^\mu \cap C_2^\nu}^{KL}(\mathbf{K}_\theta)$ of $\mathbf{K}_\theta$ on the constraint set $C_1^\mu \cap C_2^\nu$. In this case, we can write an iterative algorithm for robust alignment called GCA-RINCE as follows:

$$L_{\text{GCA-RINCE}}^{\lambda,q} = \min_\theta -q^{-1}(\text{diag}(\mathbf{P}_\theta^{(2t-1)})/\mathbf{u}^{(t)})^q + q^{-1}(\lambda \mathbf{P}_{tgt}/\mathbf{u}^{(t)})^q, \tag{13}$$

where $\lambda$ and $q$ are hyperparameters, $\mathbf{P}^{(1)} := \text{diag}(\mathbf{u}^{(1)})\mathbf{K}_\theta \text{diag}(\mathbf{v}^{(0)})$, and $t$ is the number of iterations.

## 4.3 Connection to BYOL

Our framework also allows us to make connections to BYOL [22]. BYOL learns by encouraging similarity between positive image pairs, without explicitly conditioning on negative examples. To build this connection, recall that BYOL has the online network parameterized by $\theta$ and target network parameterized by $\xi$, where $\mathbf{z}_\theta' = \widetilde{f}_\theta(\mathbf{x}')$ and $\mathbf{z}_\xi'' = \widetilde{f}_\xi(\mathbf{x}'')$ are the normalized outputs of the online and target networks, respectively. A simplified version of the BYOL loss can be written as: $L_{\text{BYOL}} = \|\widetilde{q}_\theta(\mathbf{z}_\theta') - \mathbf{z}_\xi''\|_2^2$, where $\widetilde{q}_\theta(\mathbf{z}_\theta')$ is the normalized output after online network and $q_\theta$ is the predictor.[3] In this case, we can provide the following connection between GCA and BYOL as follows.

**Theorem 4** (BYOL Equivalence). *Let $\mathbf{S}_\theta(\mathbf{x}_i', \mathbf{x}_j'') = \exp(-\|\widetilde{q}_\theta(\mathbf{z}_i') - \mathbf{z}_j''\|)$ denote the augmentation kernel. Set the target plan $\mathbf{P}_{tgt} = \mathbf{I}$, $d_\Gamma$ equal to the L2-distance, $d_M$ equal to the KL-divergence, and constraint set as $R^{B \times B}$. The BYOL objective can be re-expressed as a GCA problem as follows:*

$$\min_\theta KL\big(\mathbf{I}||\mathbf{S}_\theta\big), \quad \text{with} \ \ \mathbf{S}_\theta = Prox_{R^{B \times B}}^{\|\cdot\|}(\mathbf{S}_\theta). \tag{14}$$

See the proof in Appendix B.6.

---

[3]In practice, BYOL also switches the order of views to symmetrize the loss. For ease of discussion, we consider just one pair of views but the same could be argued for the full symmetric version.

# 5 Theoretical Analysis

In this section, we aim to show how the GCA-methods can improve alignment and uniformity in the latent space [57]. Here, *alignment* means that the features of the positive samples are as close as possible, while *uniformity* means that the features of negative samples are uniformly distributed on latent space (see Appendix C.1 for formal definitions). These quantities have been studied in a number of related works [57, 45], where one can show that improved alignment and uniformity can lead to different benefits in representation learning.

## 5.1 Improved alignment with GCA

Contrastive learning minimizes the deviation between the target alignment plan with the transport plan in Definition 3 through empirical risk minimization (ERM). Therefore, a tighter bound on the empirical risk corresponds to a smaller difference between the ideal alignment with the coupling matrix. We show that this in turn leads to better alignment of the positive views.

**Analysis of INCE vs GCA-INCE.** GCA-INCE ensures that the final transport plan $\mathbf{P}^{(\infty)}$ is closer to the ideal identity matrix compared to the INCE, as we show in the following theorem.

**Theorem 5** (Improved Alignment with INCE). *Let $\mathbf{K}_\theta$ denote the augmentation kernel as in Definition (3). Set $d_M$ and $d_\Gamma$ to the KL-divergence, and $\mathbf{P}_{tgt} = \mathbf{I}$. The GCA-INCE loss with converged plan $\mathbf{P}_\theta^{(\infty)}$ is lower than the GCA-INCE loss with $\mathbf{P}_\theta^{(t)}$ in Equation (6) for all $t$.*

The full proof is provided in Appendix C.1.1. The above theorem tells us that solving Equation (8) with iterative projection will converge to a transport plans $\mathbf{P}_\theta^{(\infty)}$ with lower KL divergence than the one-step solution provided by INCE. We can establish the convergence of the $\mathbf{P}^{(t)} \to \mathbf{P}^{(\infty)}$, based on the convergence of Bregman projection.

**Analysis of RINCE vs GCA-RINCE.** GCA also benefits from other Bregman divergences, like the WDM in RINCE, which provides robustness against distribution shift compared to the KL-divergence in INCE. GCA-RINCE provides a lower bound on the RINCE loss in Equation (2), which allows us to develop a tighter bound with $\mathbf{P}^{(\infty)}$ obtained by several proximal steps with GCA.

**Theorem 6** (Improved Alignment with RINCE). *GCA-RINCE loss with $\mathbf{P}_\theta^{(t)}$ in Equation (13) is lower than the loss in the Theorem (2) as $L_{GCA\text{-}RINCE}^{\lambda, q=1}(\mathbf{P}_\theta^{(t)}) \leq L_{RINCE}^{\lambda, q=1}(\mathbf{P}_\theta^{(1)})$.*

See Appendix C.1.1 for the full proof and an analysis of GCA methods for different choices of $d_M$.

## 5.2 Improved Uniformity of Representations Through GCA

The improved alignment of GCA-methods comes from maximization of the uniformity under the constraint of intersection $C_1^\mu \cap C_2^\nu$ in Equation (4), rather than the constraint set $C_1^\mu$ in INCE (see Table 1). Finding the projection of $\mathbf{K}_\theta$ on set of $C_1^\mu \cap C_2^\nu$ through proximal steps is equivalent to solving the dual problem of EOT, which can be summarized through the following theorem.

**Theorem 7** (Improved Uniformity). *Given the constraint sets in Equation (4), the optimal transport coupling upon convergence of Equation (6), denoted as $\mathbf{P}^{(\infty)}$, achieves a higher uniformity loss compared to the single-step transport plan $\mathbf{P}^{(1)}$ obtained by INCE.*

The proof is provided in the Appendix C.2. Through loss propagation, we show that the alignment plan offered by $\mathbf{P}^{(\infty)}$ will guide the subsequent iterations towards more uniform representations.

## 5.3 Impacts of GCA on a downstream classification task

We take this one step further and examine the impact of GCA on a downstream classification task. For a classification task, using a labeled dataset $\mathcal{D} = \{(\bar{\mathbf{x}}_i, \mathbf{y}_i)\} \in \bar{\mathcal{X}} \times \mathcal{Y}$ where $\mathcal{Y} = [1, \ldots, M]$ with $M$ classes, we consider a fixed, pre-trained encoder $f_\theta \in \mathcal{F} : \mathcal{X} \to \mathcal{S}$. Assume that positive and negative views of $n$ original samples $(\bar{\mathbf{x}}_i)_{i \in [1..n]} \subset \bar{\mathcal{X}}$ are sampled from the data distribution $p(\bar{\mathbf{x}})$.

In this case, the uniformity loss is equivalent to optimizing the downstream supervised classification tasks with cross-entropy (CE) loss when the following two assumptions are satisfied [16].

**Assumption 1** (Expressivity of the Encoder). *Let us define $\mathcal{H}_{\bar{\mathcal{X}}}$ is the RKHS associated with the kernel $\mathbf{K}_{\bar{\mathcal{X}}}$ defined on $\bar{\mathcal{X}}$, and $(\mathcal{H}_{f_\theta}, \mathbf{K}_\theta)$ defined on $\mathcal{X}$ with augmentation kernel $\mathbf{K}_\theta = \langle f_\theta(\cdot), f_\theta(\cdot) \rangle_{\mathbb{R}^d}$ in Definition 3. And we assume that $\forall g \in \mathcal{H}_{f_\theta}, \mathbb{E}_{\mathcal{A}(x|\cdot)} g(x) \in \mathcal{H}_{\bar{\mathcal{X}}}$.*

| | Standard Setting | | | | Noisy Setting | | | |
|---|---|---|---|---|---|---|---|---|
| Method | CIFAR-10 | CIFAR-100 | SVHN | ImageNet100 | CIFAR-10 (Ex) | CIFAR-100 (Ex) | CIFAR-10C | CIFAR-10C (Ex) |
| INCE | 92.01 ± 0.40 | 70.07 ± 0.42 | 90.60 ± 0.17 | 73.01 ± 0.61 | 82.03 ± 0.32 | 54.70 ± 0.43 | 87.20 ± 0.37 | 74.84 ± 0.21 |
| GCA-INCE | _92.36 ± 0.24_ | _70.11 ± 0.45_ | 90.40 ± 0.16 | 73.04 ± 0.76 | 82.18 ± 0.69 | 54.91 ± 0.56 | 87.34 ± 0.34 | 76.00 ± 0.17 |
| Δ | +0.35 | +0.04 | -0.20 | +0.03 | +0.15 | +0.21 | +0.14 | +1.16 |
| RINCE | 91.05 ± 0.50 | 69.06 ± 0.64 | 90.97 ± 0.19 | 71.91 ± 0.43 | 82.60 ± 0.63 | 55.43 ± 0.48 | 88.62 ± 1.33 | 77.05 ± 0.82 |
| GCA-RINCE | 92.09 ± 0.22 | 69.72 ± 0.27 | _91.45 ± 0.41_ | _73.44 ± 0.55_ | _82.76 ± 0.49_ | _55.90 ± 0.41_ | _88.76 ± 0.72_ | _77.23 ± 0.76_ |
| Δ | +1.04 | +0.66 | +0.48 | +1.53 | +0.16 | +0.47 | +0.14 | +0.18 |
| SimCLR | 92.16 ± 0.16 | 69.95 ± 0.14 | 90.24 ± 0.24 | 72.20 ± 0.78 | 81.87 ± 0.53 | 54.54 ± 0.79 | 86.98 ± 1.59 | 73.79 ± 0.32 |
| BYOL | 90.56 ± 0.59 | 69.75 ± 0.37 | 89.50 ± 0.46 | 69.75 ± 0.83 | 81.55 ± 0.50 | 54.18 ± 0.46 | 87.88 ± 1.02 | 69.40 ± 1.11 |
| IOT [51] | 90.99 ± 0.54 | 67.19 ± 0.21 | 90.15 ± 0.21 | 72.27 ± 0.53 | 80.59 ± 0.64 | 52.40 ± 0.48 | 67.36 ± 1.97 | 58.75 ± 1.96 |
| IOT-uni [51] | 90.89 ± 0.57 | 67.03 ± 0.40 | 90.54 ± 0.20 | 72.88 ± 0.71 | 80.79 ± 0.24 | 53.04 ± 0.52 | 69.58 ± 1.25 | 59.05 ± 1.86 |
| GCA-UOT | **92.61 ± 0.32** | **71.45 ± 0.37** | **91.96 ± 0.15** | **74.09 ± 0.40** | **83.18 ± 0.44** | **56.30 ± 0.51** | **89.61 ± 0.30** | **77.60 ± 0.54** |

Table 2: *Test accuracy (%) on a downstream classification task after pretraining.* Results are provided for CIFAR-10 (ResNet18), CIFAR-100 (ResNet18), SVHN (ResNet50), and ImageNet100 (ResNet50) under standard and extreme (Ex) augmentation conditions (averaged over 5 seeds). The top model is bold and the second-place model is underlined. For INCE and RINCE, we also provide the improvement Δ by adding GCA to each method.

**Assumption 2** (Small Intra-Class Variance). *For $y \neq y'$, the intra-class variance $\delta_i, \delta_j$ are negligible compared to the distance among different class centroids, $\mu_y, \mu_{y'}$ as $\|\mu_y - \mu_{y'}\| \gg \|\delta_i - \delta_j\|$.*

**Claim 1.** *If Assumption 1 and Assumption 2 hold, then maximizing the uniformity is equivalent to minimizing the downstream CE loss.*

The proof is provided in Appendix C.2. Optimizing the self-supervised loss under ideal conditions improves downstream CE tasks and helps to explain why maximizing uniformity aids classification.

**Remark.**. Maximizing uniformity can enhance downstream classification but risks "feature suppression" by encouraging shortcut features that harm generalization [48]. In GCA-UOT, adding penalties modifies the transport plan from that of a pure uniformity loss, helping to avoid feature suppression. We find empirical evidence that UOT provides a more robust transport plan which appears to circumvent some of these shortcut features from being learned (Figure A4 in Appendix C.3).

# 6 Experiments

In this section, we conduct empirical evaluations to study the performance of our approach in both handling noisy and corrupted views and in domain generalization tasks.

## 6.1 Comparison with CL Baselines

**Experiment setup.** To examine the robustness of our framework, we trained INCE and RINCE as baselines, and developed their GCA-based alternatives (+GCA). In addition, we also compared with our novel GCA-UOT method, two variants of IOT established in [51], and other CL baselines, including BYOL and SimCLR. For experiments with SVHN [36] and ImageNet100 [15] we use the ResNet-50 encoder as the backbone and use a ResNet-18 encoder as the backbone for CIFAR-10, CIFAR-100 [29] and a corrupted version of CIFAR called CIFAR-10C [25].

In all of these cases, we follow the standard self-supervised learning evaluation protocol [8], where we train the encoder on the training set in an unsupervised manner and then train a linear layer on top of the frozen representations to obtain the final accuracy on the test set. In addition to standard data augmentation policies commonly used [12], we also apply three different extreme augmentation policies to examine the robustness of GCA towards noisy views (details in Appendix D.2). Learning rates and other training details for CIFAR-10, CIFAR-100, SVHN, and ImageNet100 are provided in Appendix D.1, while specific training details for CIFAR-10C are included in Appendix D.2.

**Results on Standard Augmentations.** First, we performed experiments on CIFAR-10, CIFAR-100, SVHN, and ImageNet100 using standard sets of augmentations that are applied to achieve state-of-the-art performance (Table 2, Standard Setting). We found the +GCA versions of INCE and RINCE exhibit performance gains in almost all settings except for SVHN, with bigger gains observed when adding GCA to RINCE. Additionally, we find that our unbalanced OT method, GCA-UOT, achieves the top performance across the board, on all four datasets tested. The transport plans obtained by each methods are provided in Figure A4 along with a study of the sensitivity of the methods to hyperparameters (Appendix A7).

**Results on Corrupted Data and Extreme Augmentations.** Next, we tested the methods in two noisy settings. In the first set of experiments, we apply extreme augmentations to CIFAR-10 (Ex) and

CIFAR-100 (Ex) (see Appendix D.2) to introduce noisy views during training. In the second set of experiments, we used the CIFAR-10C to further test the ability of our method to work in noisy settings. Our experimental results demonstrate that the GCA-based strategy effectively enhances the model's generalization ability and adaptability to aggressive data augmentations. In addition to improving classification accuracy, the GCA-based methods also improve the representational alignment and uniformity, as shown in Appendix E.2. This observation is in line with our theoretical analysis in Section 5.2, where we show that the obtained representations provide better overall alignment of positive views and better spread in terms of uniformity [57].

## 6.2 Block Diagonal Transport in Domain Generalization

In a final experiment, we aimed to demonstrate the flexibility and robustness of our framework by applying it to a domain generalization task, where samples originate from different domains (e.g., Photo, Cartoon, Sketch, Art). We explored the effects of introducing domain-specific alignment constraints in our transport plan, hypothesizing that this could enhance the latent space organization to capture more nuanced domain similarities.

Our approach enables additional contextual information to be seamlessly integrated into the transport process. In this case, domain information was incorporated to distinguish the alignment of samples from the same versus different domains. To achieve this, we adjusted the target transport plan $\mathbf{P}_{tgt}$, selectively modifying parameters $(\alpha, \beta)$ to vary the influence of domain-based alignment constraints as shown in Figure 1(A). Specifically, we set $\{\alpha = 0, \beta > 0\}$ to prioritize cross-domain alignment and $\{\alpha > 0, \beta = 0\}$ to focus on intra-domain alignment.

The training was conducted on the PACS dataset [31] using a ResNet-18 encoder with the GCA-INCE objective. After training the encoder in an unsupervised manner, we freeze the encoder and then train a linear readout layer to predict either the sample's class or the domain it belonged to. This setup allowed us to isolate the effect of our transport adjustments on the latent space's capacity to encode both class and domain information.

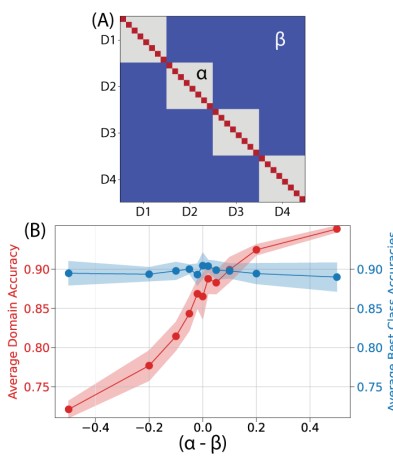

Figure 1: *Incorporating different priors into learning across multiple domains.* (A) Example target alignment plan $\mathbf{P}_{tgt}$, where the target over all samples from the same domain are set to $\alpha$, the diagonal values are set to 1, and across-domain samples are set to $\beta$. (B) The domain classification accuracy (red) and overall class accuracy (blue) with $(\alpha - \beta)$ increases.

The results, displayed in Figure 1(B), revealed that increasing the domain alignment weight enhances the accuracy of domain classification (from 72.11% to 95.16%) without diminishing classification performance. This outcome suggests that GCA can effectively encode both domain and class information in a single latent representation. The ability to adjust alignment constraints provides a powerful tool for domain generalization tasks, enabling multiple types of similarity to be jointly encoded. This flexibility can potentially alleviate issues related to information loss from data augmentation, especially in fine-grained classification settings, by retaining essential domain-specific characteristics across transformations.

## 7  Conclusion

In this work, we introduced *generalized contrastive alignment* (GCA), a flexible framework that redefines contrastive learning as a distributional alignment problem using optimal transport to control alignment. By allowing targeted control over alignment objectives, GCA demonstrates strong performance across both standard and challenging settings, such as noisy views and domain generalization tasks. This work opens up broader possibilities for learning robust representations in real-world scenarios, where data is often diverse, noisy, or comes from multiple domains.

Future work includes applications of GCA to graphs and time series data, as well as multi-modal settings where our approach can integrate various forms of similarity. As alignment strategies become integral to contrastive learning, GCA offers a promising foundation for more adaptive and expressive self-supervised models.

## Acknowledgements

We would like to thank Mehdi Azabou, Divyansha, Vinam Arora, Shivashriganesh Mahato, and Ian Knight for their valuable feedback on the work. This work was funded through NSF IIS-2212182, NSF IIS-2039741, and the support from the Canadian Institute for Advanced Research (CIFAR). We would also like to acknowledge the use of ChatGPT for providing useful feedback and suggestions on the writing of the paper.

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
