# OpenReview forum: "Your contrastive learning problem is secretly a distribution alignment problem"
_NeurIPS.cc/2024/Conference — NeurIPS 2024 poster_

### Official Review · Reviewer_VXuY · 2024-07-03

**Soundness:** 4
**Presentation:** 3
**Contribution:** 4
**Rating:** 6
**Confidence:** 4

**Summary:**

The paper presents a novel perspective on contrastive learning (CL) by framing it as a distribution alignment problem using entropic optimal transport (OT). It trains an encoder network $f_\theta$ by iteratively updating encoder parameter $\theta$ and corresponding transport plan $P$ among encoded augmentations of samples. The authors establish connections between noise contrastive estimation losses widely used in CL and distribution alignment with OT. This novel connection allows for the development of various loss functions and multi-step variants for existing CL methods. The theoretical insights and experimental evidence provided demonstrate the benefits of this approach in improving the generalization and robustness of contrastive alignment in both clean and noisy settings.

**Strengths:**

The paper offers a fresh view on contrastive learning by linking it to optimal transport, providing a solid theoretical foundation for understanding and improving CL methods. The authors provide rigorous theoretical analysis and proofs for the convergence of their proposed methods, offering strong support for their claims. The proposed Generalized Contrastive Alignment (GCA) framework is versatile, allowing for the incorporation of domain-specific knowledge and customization of representation spaces.

Originality

The paper presents an innovative connection between contrastive learning and optimal transport, a novel perspective that has not been extensively explored before. The introduction of the Generalized Contrastive Alignment (GCA) framework offers a fresh approach to enhancing contrastive learning methods.

Quality

Theoretical insights are rigorously developed, providing a solid foundation for the proposed connections and methodologies. The experiments demonstrate the benefits of the GCA approach from a few different perspectives.

Clarity

The paper is logically structured, and key concepts and methods are clearly explained.

Significance

By bridging the gap between contrastive learning and optimal transport, the paper opens up new prospects for research and application in self-supervised learning. The GCA framework has the potential to improve the expressiveness and robustness of representations in various domain generalization settings, which is relevant for real-world applications.

**Weaknesses:**

While the theoretical convergence of the algorithm is established, the specific criteria for convergence are not clearly defined. The paper's use of proximal operators for T steps and auto-differentiation for parameter θ could impose significant computational burdens, especially with large T. This potential issue is not sufficiently addressed, raising concerns about the algorithm's efficiency in large-scale applications.The experiments lack a detailed runtime analysis, leaving the computational efficiency of the algorithm unexamined. The computational resources necessary for implementing the algorithm are not clearly outlined, raising concerns about its scalability and practicality for large-scale applications.

**Questions:**

In your experiments, you utilize a fixed number of epochs. How do you ensure that the convergence of the algorithm is achieved within this fixed number of iterations? Would it be feasible to implement a convergence criterion to replace the fixed number of iterations?

Optimal transport is known to require substantial computing resources, especially with large sample sizes. Could you provide a runtime analysis for your algorithm? Does the algorithm face significant computational challenges when dealing with large datasets?

You utilized proximal operators for T steps to obtain the transport plan P, with the differentiation of the parameter θ computed via auto-differentiation from the loss, propagating back through T steps. Could the auto-differentiation process impose a significant computational burden when a large number of steps T is required in the proximal operator? How do you mitigate this potential issue?

Would you method be able to handle more than 2 types of augmentation? What would be your target transport plan if more than 2 augmentation is used in the contrastive learning?

---

> ### Author Rebuttal · Authors · 2024-08-07
>
> Thank you for your feedback. We appreciate your time and suggestions. Our specific responses to your questions are provided below.
>
>
> 1. “ the specific criteria for convergence are not clearly defined. [...] concerns about the algorithm's efficiency in large-scale applications. The experiments lack a detailed runtime analysis,[...]..”
>
> **Reply:**  Thanks for your questions. We summarize our computational analysis in line Line 188 in our main text and give the run time analysis in Sec. 3.1. We show the cost of MSINCE in 10 iterations is only 5% more operations in Flops than INCE, and GCA-UOT is even lower than the INCE with a  30% reduction in Flops in general response (#2).
>
>
> 2. “ How do you ensure that the convergence of the algorithm is achieved within this fixed number of iterations? Would it be feasible to implement a convergence criterion to replace the fixed number of iterations?”
>
> **Reply:**  Thanks for your questions. In practice, we use a simple convergence criteria for automatically terminating the multistep algorithm. For CIFAR-10, we found that we could also set the maximum iterations to 5 without any loss in performance.
> Based upon the reviewers comments, we ran an experiment to examine the impact of the number of iterations on the accuracy and compactness of the classes (Fig. U2). We found that the accuracy was not very sensitive to the exact choice for the number of fixed iterations with comparable performance for anywhere from 5-11 iterations.
>
> 3. “Optimal transport is known to require substantial computing resources, especially with large sample sizes. Could you provide a runtime analysis for your algorithm? Does the algorithm face significant computational challenges when dealing with large datasets?”
>
> **Reply:**  We provide an analysis of computational complexity and empirical results in the Appendix in Sec. C.1. The computation efficiency is also discussed in our general response, where we show the small amount of overhead for MS-INCE (5% more flops) and a 30% reduction in flops and running time for GCA-UOT.
>
> In terms of dealing with larger datasets, please see our results on ImageNet-100 and SVHN in the general response (Table R1). There we show that the model scales to larger datasets and still performs on par with baseline methods.
>
> 4. “You utilized proximal operators for T steps to obtain the transport plan P, with the differentiation of the parameter θ computed via auto-differentiation from the loss, propagating back through T steps. Could the auto-differentiation process impose a significant computational burden when a large number of steps T is required in the proximal operator? How do you mitigate this potential issue?”
>
> **Reply:** Thank you for your question. We don’t backpropagate the loss at different iterations in our alignment objective. Instead, we compute a final transport plan, and then backpropagate the loss, similar to the way BatchNormalization operates. While the number of T steps could become substantial when dealing with large datasets and small mini batch, we have implemented several strategies to address this:
> - We perform the optimal transport (OT) computation in the latent space rather than the input space, which reduces the data's dimensionality involved in the OT problem, thereby significantly decreasing both computational cost and memory requirements.
>
> - We set a convergence threshold to prevent an excessively large number of T steps. This ensures that the algorithm stops as soon as an adequate solution is found, further optimizing computational efficiency.
>
> 5. “Would you be able to handle more than 2 types of augmentation? What would be your target transport plan if more than 2 augmentation is used in contrastive learning?”
>
> **Reply:**   By having more than two types of augmentations, do you mean having multiple views of the same example in the same batch? If so, then yes, we could handle that case with our framework! We show how this many-to-one matching can be implemented as a block diagonal constraint on the transport plan in our domain generalization experiments.
> We hope to include this as an example for future work. Thanks for your suggestion.

---

> > ### Comment · Reviewer_VXuY · 2024-08-10
> >
> > Thanks for the author's reply and the detailed runtime analysis provided. This has alleviated my concerns about the practical applicability of the proposed algorithm.

---

> ### Author Response · Authors · 2024-08-13
>
> Dear Reviewer VXuY,
>
> We greatly appreciate your constructive comments and are glad that the additional analysis addressed the issues you raised. We kindly hope that you might consider increasing your score based upon the discussion.

---

### Official Review · Reviewer_TBTc · 2024-07-11

**Soundness:** 4
**Presentation:** 1
**Contribution:** 3
**Rating:** 4
**Confidence:** 4

**Summary:**

The paper recasts several self-supervised learning (SSL) paradigms, such as SimCLR, in an optimal transport framework. In many SSL variants each batch contains two views of the same data sample. This means that the embeddings of a batch can be viewed as the union of two sets, each one containing only one of the two views of each data sample. The key insight of the paper is that the typical SSL losses can be viewed as the discrepancy between approximations of the optimal transport plans between these two sets from the target transport plan that simply matches the two views of each sample. From this OT perspective the authors propose variants of self-supervised learning losses, improving the approximation to the optimal transport plan, generalizing to weighted batches, and relaxing the target transport plan to incorporate prior domain knowledge.

**Strengths:**

Originality
- The connection between SSL and OT discussed in this paper seems to extend prior works on the relation of SSL and OT.

Quality
- Connecting two active areas of research, such as SSL and OT, is very useful.
- Phrasing SSL objectives in an OT framework provides an interesting new perspective on what SSL does.

Clarity:
- The authors provide a dedicated appendix elaborating the concept of proximal operators, which is helpful as this is more of a niche topic for the general ML community.

Soundness:
- Their reformulation of SSL losses is correct and most statements are precisely formulated.

**Weaknesses:**

I appreciate the new perspective the paper presents a lot. However, the presentation needs to be significantly improved.

- *W1 Presentation*
	- To me the motivation for introducing all the notation and machinery in sections 2.2-3.3 was not sufficiently clear when reading the paper for the first time. Many abstract concepts are introduced in great generality (lots of choices for divergences, constraint sets etc), but the first real benefit of using this framework appears only on page 6 (Thm 1). I would suggest to drastically restructure the paper and use recasting INCE in the OT framework as the red thread. Flesh out the proof of Thm 1 by explaining how the term inside the logarithm of INCE can be seen as half a Sinkhorn step and use this to motivate the introduction of proximal operators. Then explain how one gets the full INCE loss when using the KL divergence between the approximately optimal transport and the target transport plan. This way every ingredient of the OT framework is directly motivated. Once the INCE case is discussed you can generalize the OT perspective to also incorporate other SSL losses and generalize them.
	- My understanding is that the terms "half-Sinkhorn step", "Bregman projection", and "proximal operator" are pretty much synonyms, at least in the context of this work. If so, I would recommend to stick to one of the three terms (while possibly maintaining an appendix section that explains their relations). This would make the exposition much more accessible. My preference would be "half-Sinkhorn step" as Sinkhorn operations are most widely known in the ML community. I would even suggest simplifying the terminology at the expense of generality (maybe the BYOL connection really needs proximal operators rather than half-Sinkhorn steps) and defer the fully general setting (explaining the BYOL connection) to the appendix if needed.
	- Overall, I would recommend to introduce as little jargon as necessary to present the results. The most general setting can be discussed later in the paper or in the appendix.
	- I recommend phrasing the main idea more clearly early on in the paper (similar to my summary). In particular, lines 30-32 made me think that the transport plans are restricted to only match positive pairs. But instead they are only penalized if they do not. I only understand the setup correctly on page 4.
	- The notation used in the main paper needs to be properly introduced in the main paper. For instance, what do the subscripts $\mathcal{F}$ and $\mathcal{G}$ stand for in line 199? Define what the target coupling plan $P_{tgt}$ is supposed to do in line 167. What is $R^{B\times B}$ in line 265? If $R$ stands for the real numbers, rather use $\mathbb{R}$.

- *W2* There is no related work section. It would be useful to at least explain in which ways the present work extends prior works on the connection between SSL and OT, such as [39]

**Minor:**
- *W3* Line 107:  The equation given for the cost is nearly the cosine similarity (but not quite due to the absolute value) but the sentence states that the cost often encodes the L2 distance. This is confusing as the provided formula does not encode the L2-distance.
- *W4* Several cross-reference links are faulty. For instance, the links to Appendix A.4 and algorithm A1 in lines 188, 189 all point to algorithm 1.
- *W5* Combine Thm 7 and 6 into one as Thm 7 is a strict generalization of 6.
- *W6* The INCE result on CIFAR-10 is unusually low. Other sources (Damrich et al. 2023 or https://github.com/p3i0t/SimCLR-CIFAR10) report linear classifier accurarcies above 92% for SimCLR on Cifar-10, much higher than for any SSL variant in this paper.
- *W7* Casting BYOL into the GCA framework seems pretty forced: The proximal operator is the identity, the result $S_\theta$ is really not a transport plan, and applying the KL loss seems odd as there is no normalization constraint on $S_\theta$. I think the ultimate reason for this less on the GCA side and has more to do with the lack of a repulsive force in BYOL (and the existence of a degenerate optimum with a constant encoder). I would perhaps recommend deferring this finding to the appendix, which might also allow to reduce the level of generality required in the main paper, see above.

**Questions:**

- *Q1* There seem to be multiple ways for enforcing / penalizing constraints: In eq (8) the valid choices for $P$ are restricted both by $\mathcal{B}$ and $h(P)$, especially if the latter is an indicator function. Why use both ways of representing constraints and not include both either in $\mathcal{B}$ or $h$? Similarly in eq (9) there are additional divergences to achieve non-uniform marginals of the transport plan. Could this not also be subsumed by $h$ or $\mathcal{B}$?

- *Q2* The convergence proofs mentioned in line 59, line 188 and A4 only refer to the inner optimization loop of finding the optimal transport plan, right? In particular, they do not show that the parameters of the neural network will converge independent of the data. If so, rephrase the statements in lines 59 and 188 accordingly. Currently, they can be misunderstood as an overstatement.

- *Q3* In the full GCA setup, where the optimal transportation plan is computed (not just one step), the backwards pass needs to unroll all the Sinkhorn steps, right? Does this not lead to high complexity and potentially exploding / vanishing gradients? In particular, I was surprised by line 186 stating that the forward-pass does not affect the computational complexity of the backward pass.

- *Q4* How are the marginal distributions $\mu$ and $\nu$ chosen for unbalanced GCA in section 6.2 and Table 2?

- *Q5* What is the value of $\mu$ in Thm 1, 2, and 3? Is is simply the vector of all ones?

- *Q6* While the statements in Theorem 5- 8 are interesting, I wonder why having higher / lower losses should help with better representations?. The authors do show this empirically, but I wonder why one would expect this from just from having higher / lower losses. Similarly, I wonder why higher uniformity loss implies improved uniformity (Thm 5) and why lower general loss implies improved alignment (Thm 6). For instance, could one not argue that since the loss is already lower, there is less learning signal and thus worse alignment? Also it is the full GCA loss that is lower than the full INCE loss in Thm 6 not just the attractive (alignment) part.

- *Q7* What is the unit for the y-axis in Fig A3? Is it seconds?

**Limitations:**

Limitations are not explicitly discussed, but I also do not see obvious limitations that need to be stated.

---

> ### Author Rebuttal · Authors · 2024-08-07
>
> We sincerely thank you for your thorough evaluation and insightful feedback on our manuscript. Based upon your suggestions, we plan to make major revisions to this paper to improve the quality of the presentation. Below we provide replies to your other questions and concerns.
>
> 1. Suggestions on restructuring the paper
>
> **Reply:**  We sincerely thank you for the suggestions to rearrange our paper. We thought long and hard about this and originally started with INCE and Sinkhorn in a previous version. However, we decided to go with this organization as it sets up the problem in full generality before diving into specific examples of the idea in action.
>
> 2. Terminology surrounding alignment iterations
>
> **Reply:**  Since the use of the Sinkhorn algorithm is a specific case in our framework, only applied when using INCE, we have referred to it as a "half-Sinkhorn" step. As we extend to more general divergences and losses, proximal operators provide a more generalized framework for this optimal transport approach. We will simplify our terminology and incorporate your ideas to clarify our motivation. We appreciate your detailed suggestions!
>
> 3. “What do the subscripts $\mathcal{F}$ and $\mathcal{G}$ in line 199  [..] what the target coupling plan Ptgt in line 167 [...]. What is $R^{BxB}$ in line 265 [...]?”
>
> **Reply:**  Thank you. In line 199, $\mathcal{F}$ and $\mathcal{G}$ are constraint sets for divergences $h_F$ and $h_G$. In line 167, $P_{\text{tgt}}$ is the target distribution, set as the identity matrix (line 171) or other relaxed forms (line 208). In line 265, $\mathbb{R}^{B \times B}$ is correct. We will clarify these terms in our revision.
>
> 4. Discussion of related work
>
> **Reply:**   Thanks. We added some related work in the background section but agree that a more thorough discussion is needed. We plan to expand this and clarify the differences from [39] in our revision.
>
> 5. “Line 107: The equation given for the cost [...]”
>
> **Reply:**  Thanks. It was a typo in line 107, and we've corrected it to cosine similarity.
>
> 6. “Combine Thm 7 and 6 [...]”
>
> **Reply:**  Thanks, we agree with your suggestion. We plan to combine the two theorems as you suggested.
>
> 7. “The INCE result on CIFAR-10 is unusually low.[...]”
>
> **Reply:** Thanks. Our results are based on 400 epochs with an SGD optimizer for the linear layer. After training for 1000 epochs with a LARS optimizer, we achieved 90.42% accuracy. Our approach only considers pairwise matching across B samples, not the full SimCLR implementation with 2B samples, causing a slight decrease in performance compared to reported SimCLR results.
>
> 8. “Casting BYOL into the GCA framework seems pretty forced. [...] ”
>
> **Reply:**  Thanks. While our formulation may not be the most natural for expressing BYOL, we believe BYOL demonstrates the flexibility of our proximal theory framework by simply changing the kernel \(\bf K_\theta\). We will consider your suggestion for the final submission.
>
> 9. “Why use both ways of representing constraints and not include both either in B or h?”
>
> **Reply:**  Thanks. The set $\mathcal{B}$ is the set of feasible solutions, while $h(P)$ encodes the penalty. Thus, finding the solution for $\mathbf{P}$ is influenced by both. In Eq. (9), we show that constraints in $\mathcal{B}$ can be converted into a soft penalty via functions $h_i$ (where $i = 1, 2, \ldots, n$) by finding their dual formulation.
>
> 10. “The convergence proofs only refer to the inner optimization loop [...] rephrase the statements in lines 59 and 188 accordingly. ”
>
> **Reply:** Thanks. Yes, it only refers to the inner loop. We will rephrase the statements in lines 59 and 188 to make this clear. Our theory for this is detailed in Theorems 5-8.
>
> 11. “The backwards pass needs to unroll all the Sinkhorn steps, right?” [...] “Does this not lead to potentially exploding/vanishing gradients”
>
>
> **Reply:**  No, the backward pass doesn't need to unroll all the Sinkhorn steps (or iterations for general losses). See our general response (#2) on complexity. As shown in Fig. A3, the computational resources for the backward pass aren't significantly affected by the number of iterations. Regarding the impact on gradients, incorrect parameters like epsilon could influence them. This can be mitigated by selecting appropriate regularization parameters.
>
> 12.  “How are the marginal distributions $\mu$ and $\nu$ are chosen for unbalanced GCA”
>
> **Reply:**  In Table 2, the target transport plan for GCA-UOT is set to the identity matrix. In Sec. 6.2, we explore different constraints by varying the alpha and beta values in the target transport plan matrix, so the values of $\mu$ and $\nu$ change with it.
>
> 13. “What is the value of \mu” in Thm 1, 2, and 3? Is it simply the vector of all ones?
>
> **Reply:** Yes, they are vectors of ones. We will clarify this in the final version.
>
> 14. “[...] why having higher / lower losses should help with better representations?. [...]”
>
> **Reply:** That's an excellent question. Proving the benefits for the lower bound is challenging due to training dynamics. In Appendix A10 and Fig. A2, we explain that INCE aligns $\bf{P}$ in one direction with $\bf{P}\bf{1}_m=\bf{a}$, while GCA aligns samples in two directions with $\bf{P}^T\bf{1}_n=\bf{b}$, leading to more uniform results. In Theorem 6, we show that our approach improves alignment loss because optimizing INCE loss is akin to optimizing alignment loss. We will clarify this in our final submission.
>
> 15. “What is the unit for the y-axis in Fig A3? Is it seconds?”
>
> **Reply:**  Yes, we have updated the units in Fig A3 to seconds.

---

> > ### Comment · Reviewer_TBTc · 2024-08-08
> >
> > Thank you for the detailed reply and the intention of addressing several of my points in the revision. Please find some follow-up questions / comments below.
> >
> > **1. Structure:** Restructuring the paper is clearly a lot of work. Nevertheless, I still think that you original plan is more friendly for a wider audience of readers. The presentation is the main weakness in my mind.
> >
> > **9. Use of $\mathcal{B}$ and $h$:** But you write in line 163 that $h$ is typically an indicator function. In line 896 you directly translate between $\mathcal{B}$ and $h$ with infinite penalty outside $\mathcal{B}$. Does *indicator function* imply infinite penalty in your setting? If so, I really do not see the difference between constraining via $\mathcal{B}$ and $h$. If you usually use other functions than indicators for $h$, perhaps rephrase line 163.
> >
> > **11. Complexity:** Yes, I see that empirically your method is similarly fast as other SSL methods. However, I do not understand why you do not need to unroll Sinkhorn operations to compute the gradients. How are gradients computed when the forward pass includes an inner optimization line in step 2 of Algorithm 1.
> >
> > **12. Value of $\mu$ and $\nu$:** Is it correct that $\mu$ and $\nu$ are the marginals of $P_{tgt}$, which in turn depends on $\alpha$ and $\beta$?
> >
> > **14. Higher alignment loss:** To be honest, I still do not understand the argument. On a high-level it sounds as if the INCE loss is less informative than the GCA loss, because the former has one and the latter two requirements on the marginals. But I cannot connect this with either Figure A2 or Appendix A10. I also do not understand what this has to do with the relative size of the loss values.
> >
> > Here are some specific questions regarding Figure A2: Are the blue / red points in the first panel the current batch or some random selection of positive pairs? Should the arrows in the second and third panel point to the blue / red points (they do not: for instance the orange arrows in the middle and the right panels are not the same). What are the arrows supposed to mean? Are there any projections to tangent spaces involved? Are the fat arrows in the middle plot contributions to a gradient? If so, of which point is this supposed to be the gradient? I do not get at all why there is a line for INCE but a plane for GCA INCE. How is this connected to the constraints on the marginals?
> >
> > You write in line 1315 that you perform supervised training. But INCE and GCA-INCE are self-supervised. You write in line 1316 that you perform PCA. Was this of the unnormalized points in high-dimensional space or of the normalized points? In either case the result are non-normalized. Why should PCA + renormalization to the sphere encode information about the structure in high-dimensional space? Why not retrain a resnet-18 with a 3D output space, so that you can directly visualize it.

---

> > > ### Author Response · Authors · 2024-08-10
> > >
> > > Thanks for your questions and rigorous review.
> > >
> > > 9. Use of $\mathcal{B}$ and $h$: But you write in line 163 that $h$ is typically an indicator function. In line 896 you directly translate between $\mathcal{B}$ and $h$ with an infinite penalty outside $\mathcal{B}$. Does indicator function imply infinite penalty in your setting? If so, I really do not see the difference between constraining via $\mathcal{B}$ and $h$. If you usually use other functions than indicators for $h$, perhaps rephrase line 163.
> > >
> > > Reply: Thank you for your question. The penalty function h can provide a hard constraint (using an indicator function) or it can be a different function that measures the deviation from the constraint. In our implementation of GCA-UOT, we use a KL-divergence for h to relax our constraints in balanced OT from a hard to a soft penalty. We will rephrase line 163 to make this clear.
> > >
> > >
> > > 11. Complexity: Yes, I see that empirically your method is similarly fast as other SSL methods. However, I do not understand why you do not need to unroll Sinkhorn operations to compute the gradients.
> > >
> > > Reply:  Our optimization process is decoupled and we can solve it through two different optimization procedures, one to compute the representations and the other to find the optimal transport plan. You can think of the Sinkhorn algorithm as a process that provides us with a static transport plan between two sets of points in the latent space, and what we are actually differentiating over is the cost matrix.
> > >
> > >
> > > 12. Value of $\mu$ and $\nu$: Is it correct that $\mu$ and $\nu$ are the marginals of $P_{tgt}$, which in turn depends on $\alpha$ and $\beta$?
> > >
> > > Reply: Yes, in the domain adaptation setting, the marginals depend on alpha and beta.
> > >
> > >
> > > 14a. Higher alignment loss: To be honest, I still do not understand the argument. On a high-level it sounds as if the INCE loss is less informative than the GCA loss, because the former has one and the latter two requirements on the marginals. But I cannot connect this with either Figure A2 or Appendix A10. I also do not understand what this has to do with the relative size of the loss values.
> > >
> > > Reply: The CL loss in Eq.(1) can be decomposed into two terms, the alignment loss and uniformity loss [1] which corresponds to the entropy $-\epsilon H(P)$ and $<P,C>$. Under the perfectly aligned condition where the alignment loss will be zero, we show that the uniformity loss in GCA will be lower than the original INCE objective. We can use this, along with recent work from [2] where they show that a tighter bound on the uniformity can benefit the downstream task like classification, to reason that lower loss can lead to provable benefits in learning.
> > >
> > >
> > > [1] Wang T, Isola P. Understanding contrastive representation learning through alignment and uniformity on the hypersphere, International conference on machine learning. PMLR, 2020: 9929-9939.
> > >
> > > [2] Dufumier B, Barbano C A, Louiset R, et al. Integrating prior knowledge in contrastive learning with kernel, International Conference on Machine Learning. PMLR, 2023: 8851-8878.
> > >
> > > 14b. Here are some specific questions regarding Figure A2: Are the blue / red points [...]Are the fat arrows in the middle plot contributions to a gradient? If so, at which point is this supposed to be the gradient?
> > >
> > > Reply:  We are sorry for the confusion in Figure A2. Our goal was to show that by imposing two constraints on the row and column spaces with GCA, we would get more uniformly distributed latents. The subspace shows the two sets of constraints.
> > >
> > >
> > > 14d. You write in line 1315 that you perform supervised training. But INCE and GCA-INCE are self-supervised.
> > >
> > > Reply: That’s a typo. Sorry, we want to say we train a linear layer for evaluating the representations.
> > >
> > > 14e. You write in line 1316 that you perform PCA. Was this of the unnormalized points in high-dimensional space or of the normalized points? In either case the results are non-normalized.
> > >
> > > Reply:   The PCA in line 1316 is for the normalized points that are mapped to the unit sphere in 3D. It may look like the points are un-normalized because you can see points on the other side of the sphere. We are sorry this figure is confusing. We will remove it from the final version of the paper as we believe it adds more confusion.

---

> > > > ### Comment · Reviewer_TBTc · 2024-08-13
> > > >
> > > > Thanks for the additional explanations!
> > > >
> > > > **Complexity:**
> > > > Unfortunately, I still do not understand your argument. By Alg 1, I thought the forward pass is as follows
> > > >
> > > > $\theta \to f_\theta, g_\theta \to K_\theta \to P_\theta \to \mathcal{L}_{GCA}$
> > > >
> > > > where the step $K_\theta \to P_\theta$ is an inner optimization, e.g., computed via Sinkhorn. To compute the gradients of $\mathcal{L}$ with respect to $\theta$, you need to backprop through the entire forward pass, including the step from $K_\theta$ to $P_\theta$. So yes, you ultimately want to differentiate the cost matrix $K_\theta$, but your loss depends on it only through the transport plan $P_\theta$. What am I missing?
> > > >
> > > >
> > > > **Lower uniformity loss:**
> > > > Thank you for providing the reference to Dufumier et al. which I recommend adding to section 5. Please note that the theorem they proved on the usefulness of lower uniformity loss is for their specific form of the uniformity loss. It might be interesting future work to derive a similar statement for your version of the uniformity loss.
> > > >
> > > > **Figure A2:**
> > > > Yes, I recommend either to omit figure A2 or to overhaul it and the writing in appendix A10 on page 38 significantly.

---

> > > > > ### Author Response · Authors · 2024-08-13
> > > > >
> > > > > Dear Reviewer TBTc,
> > > > >
> > > > > Thank you for your follow-up question. In the Sinkhorn algorithm, the transport plan $P_\theta $ is computed as:
> > > > >
> > > > > $P_\theta = \exp(f + g - C_\theta)/\epsilon$
> > > > >
> > > > > Here, $f$ and $g$ are the dual variables that are iteratively updated during the Sinkhorn algorithm, but crucially, they do not involve gradients with respect to $\theta$. The optimization process in Sinkhorn essentially involves scaling the rows and columns of  $P$ to satisfy the marginal constraints, which can be viewed as element-wise operations (scaling and exponentiation) on the cost matrix $C_\theta$.
> > > > >
> > > > > Because $P_\theta$ is computed using the fixed-point iteration of $f$ and $g$ that depend only on the current values of $ C_\theta $, the gradient backpropagation process is simplified. Specifically, the gradient of the loss with respect to the cost matrix $ C_\theta $ is the key part that needs to be differentiated, not the intermediate steps involving $ f $ and $g$. A typical workflow of these algorithms was shown in Figure 2 of [1], the gradient flow primarily involves differentiating through $ C_\theta $, which is done only once, and not through each step of the Sinkhorn iterations.
> > > > >
> > > > > This approach reduces computational complexity and avoids the need for backpropagation through every iterative update within the Sinkhorn algorithm, which might otherwise be computationally expensive.
> > > > >
> > > > > We hope this clarifies the process. Please let us know if you have any further questions! Your insights have been instrumental in improving our work, and we kindly hope you might reconsider your evaluation in light of these discussions and forthcoming changes.
> > > > >
> > > > > [1] Eisenberger M, Toker A, Leal-Taixé L, et al. A unified framework for implicit sinkhorn differentiation. Proceedings of the IEEE/CVF Conference on Computer Vision and Pattern Recognition. 2022: 509-518.

---

> > > > > > ### Comment · Reviewer_TBTc · 2024-08-14
> > > > > >
> > > > > > Thank you for the explanation! So you are performing implicit differentiation as in Eisenberger et al.? This would resolve my issue. Regarding your explanation, I still have the following question: During the fixed point iterations, you update $f$ and $g$ based on their previous values and $C_\theta$. This makes each iteration of $f$ and $g$ depend on $C_\theta$. Normal backprop would have to trace these dependencies on $\theta$ (this is what I meant by "unrolling the Sinkhorn steps"). I appreciate that it is close to the discussion deadline, so I understand if you cannot reply in time. I just wanted to make this comment to clarify my original concern.
> > > > > >
> > > > > > The rebuttal resolved several of my smaller points. My main issue, the structure of the paper, remains unaddressed and I still believe that restructuring would make the paper significantly more accessible. Therefore, I will keep my score. The other reviewers seem to not mind the presentation, however, and I would be fine with the paper being accepted.

---

> > > > > > > ### Author Response · Authors · 2024-08-14
> > > > > > >
> > > > > > > Dear Reviewer TBTc,
> > > > > > >
> > > > > > > Thanks for your agreement for our paper to be published! To clarify further, while the gradient backpropagation does trace these dependencies, the fact that $\theta$ remains unchanged during the Sinkhorn iterations ensures that the computational complexity for the model parameters does not increase.
> > > > > > >
> > > > > > > We greatly appreciate your suggestions on improving the presentation. We plan to incorporate your feedback into the final submission to enhance clarity. Given the current presentation and the improvements we plan to make, we kindly encourage you to reconsider your evaluation of the paper.

---

### Official Review · Reviewer_ih3d · 2024-07-13

**Soundness:** 2
**Presentation:** 2
**Contribution:** 2
**Rating:** 6
**Confidence:** 3

**Summary:**

This paper introduces a framework called Generalized Contrastive Alignment (GCA) that connects contrastive learning to distribution alignment using optimal transport. The key contributions include:

1. Establishing a novel class of losses and algorithms for representation learning through GCA, showing how different contrastive losses can be interpreted as variants of a generalized distribution alignment objective.

2. Proving convergence of GCA-based methods and demonstrating theoretically that the alignment objective can improve the quality of learned representations.

3. Empirically validating GCA's effectiveness in image classification and domain generalization tasks, showing it can achieve superior performance over baseline methods.

4. Demonstrating how GCA allows building unbalanced losses using tools from optimal transport, which can handle noisy views and customize representations.

5. Providing a unified framework that connects existing contrastive learning methods like InfoNCE, RINCE, and BYOL to optimal transport formulations.

6. Showing how modifying the target alignment plan in GCA can flexibly control the amount of domain knowledge incorporated into representations for domain generalization tasks.

The paper provides both theoretical analysis and experimental results to support the benefits of the GCA framework in improving representation learning and offering more flexibility compared to standard contrastive learning approaches. The authors position this work as providing new insights into the connections between self-supervised learning models and offering tools to more easily incorporate domain knowledge into learning.

**Strengths:**

This paper demonstrates several strengths across the dimensions of originality, quality, clarity, and significance:

**Originality:**
1. The paper presents a novel framework (GCA) that creatively bridges contrastive learning and optimal transport. This connection is not entirely new, but the comprehensive treatment and generalizations provided are original.
2. The formulation of contrastive learning as a distribution alignment problem offers a fresh perspective on a widely-studied topic.
3. The extension to unbalanced optimal transport and the ability to customize target transport plans are innovative additions to the contrastive learning toolkit.

**Quality:**
1. The theoretical analysis is rigorous, with clear proofs for the connections between GCA and existing methods (InfoNCE, RINCE, BYOL).
2. The empirical evaluation is comprehensive, covering multiple datasets (CIFAR-10, CIFAR-100, CIFAR-10C, PACS) and scenarios (standard classification, extreme data augmentation, domain generalization).
3. The ablation studies and comparisons against baseline methods are thorough and well-presented.

**Clarity:**
1. The paper is well-structured, with a clear progression from problem formulation to theoretical analysis and empirical validation.
2. The use of diagrams (e.g., Figure 1) helps illustrate complex concepts like customized transport plans.
3. The authors provide clear pseudocode (Algorithm 1) for the GCA method, enhancing reproducibility.

**Significance:**
1. The GCA framework provides a unifying perspective on several popular contrastive learning methods, which could facilitate further theoretical developments in the field.
2. The improved performance on classification tasks, especially under extreme data augmentation, suggests practical benefits for real-world applications.
3. The flexibility offered by customizable transport plans in domain generalization scenarios opens up new possibilities for tailoring representations to specific tasks or domains.

**Weaknesses:**

While the paper has many strengths, there are several areas where it could be improved:

1. Limited scope of empirical evaluation:
   - The experiments are primarily conducted on relatively small datasets (CIFAR-10, CIFAR-100, PACS). While these are standard benchmarks, the absence of results on larger, more complex datasets like ImageNet limits the assessment of GCA's scalability and practical impact.
   - The authors acknowledge this limitation in their conclusion, but providing some preliminary analysis or discussion on potential challenges in scaling to larger datasets would be beneficial.

2. Computational complexity:
   - The paper lacks a detailed analysis of the computational overhead introduced by the GCA framework, particularly for the multi-step variants.
   - While Algorithm 1 is provided, there's no discussion on how the additional forward passes impact training time or memory requirements compared to standard contrastive learning methods.

3. Hyperparameter sensitivity:
   - The paper introduces several new hyperparameters (e.g., number of iterations, α and β in the customized transport plan), but there's limited discussion on their impact or guidelines for setting them.
   - An ablation study or sensitivity analysis for these parameters would provide valuable insights for practitioners looking to implement GCA.

4. Comparison with other alignment-based methods:
   While the paper compares GCA with standard contrastive learning baselines, it does not compare it with other alignment-based or optimal transport-based representation learning methods (e.g., [1], [2]).
   - Such comparisons would help better contextualize the contributions of GCA within the broader landscape of alignment-based approaches.

5. Theoretical limitations:
   - The theoretical analysis, while thorough, focuses primarily on convergence and improved alignment. Discussing potential limitations or failure cases of the GCA approach would be beneficial.
   - For instance, are there scenarios where the multi-step approach might lead to worse performance or slower convergence?

6. Ablation on the number of alignment steps:
   - While the paper mentions using 5 iterations in the forward pass, there's no analysis on how performance changes with different numbers of iterations.
   - An ablation study showing the trade-off between computational cost and performance improvement as the number of iterations increases would be valuable.

7. Limited exploration of unbalanced OT:
   - While the paper introduces unbalanced OT as a potential extension, the empirical evaluation of this approach is limited.
   - More extensive experiments or analysis demonstrating the specific benefits of unbalanced OT over balanced OT in different scenarios would strengthen this contribution.

8. Clarity on practical implementation:
   - While the paper provides theoretical foundations, it could benefit from more practical guidance on implementing GCA in real-world scenarios.
   - For instance, how should practitioners choose between different variants (GCA-INCE, GCA-RINCE, GCA-UOT) for a given task?

[1] W. Wang, etal, Zero-Shot Recognition via Optimal Transport
[2] Y. Balaji, etal Normalized wasserstein distance for mixture distributions with applications in adversarial learning and domain adaptation

**Questions:**

- Can you provide a detailed analysis of the computational overhead introduced by GCA, particularly for the multi-step variants?
- How do the training time and memory usage compare to standard contrastive learning methods?
- Is there a point of diminishing returns regarding performance improvement vs. computational cost?
- What guidelines can you provide for setting the new hyperparameters introduced in GCA (e.g., number of iterations, α and β in the customized transport plan)?
- Have you observed any patterns in how these hyperparameters affect performance across different tasks or datasets?
- Can you provide an ablation study showing how performance changes with different numbers of iterations in the forward pass?
- Have you considered comparing GCA with other alignment-based or optimal transport-based representation learning methods? If so, what were the results?
- How does GCA differentiate itself from or improve upon these existing alignment-based approaches?
- Are there any scenarios in which the multi-step approach might lead to worse performance or slower convergence than single-step methods?
- Can you elaborate on any potential limitations or failure cases of the GCA approach?
- Can you provide more details or experiments demonstrating the specific benefits of unbalanced OT over balanced OT in different scenarios?
- Are there particular types of tasks or data where unbalanced OT shows the most significant improvements?
- Can you provide more concrete guidelines on how practitioners should choose between different GCA variants (GCA-INCE, GCA-RINCE, GCA-UOT) for a given task?
- Are there specific task characteristics that make one variant more suitable than others?
- In the domain generalization experiments, how sensitive is the performance to the choice of α and β in the customized transport plan?
- Have you explored using different transport plans for domains or tasks within the same dataset?
- Can you provide more details on how GCA improves robustness to noisy or extreme data augmentations compared to baseline methods?
- Are there specific types of noise or augmentations where GCA shows the most significant improvements?

**Limitations:**

The authors have not explicitly addressed limitations or potential negative societal impacts in the paper as it currently stands. While they do mention some areas for future work in the conclusion, a more comprehensive discussion of limitations and societal impacts would strengthen the paper. Here are some constructive suggestions for improvement:

1. Limitations:
   - The authors could add a dedicated "Limitations" section discussing:
     a) The current scope of experiments (e.g., limited to certain datasets and architectures)
     b) Potential computational overhead of the multi-step approach
     c) Challenges in hyperparameter tuning for the new parameters introduced
     d) Any scenarios where GCA might not be applicable or beneficial

2. Potential negative societal impacts:
   - The authors should consider adding a brief discussion on potential societal impacts, such as:
     a) Increased energy consumption due to potentially higher computational requirements
     b) Possible biases in learned representations, especially when using customized transport plans
     c) Potential misuse of the technique in privacy-sensitive applications

3. Ethical considerations:
   - A brief note on any ethical considerations in data usage or potential applications of the method would be valuable.

4. Future work:
   - Expand the current mention of future work to include specific directions for addressing identified limitations.

It's important to note that the absence of these discussions doesn't necessarily indicate oversight by the authors, but rather an opportunity to enhance the paper's comprehensiveness. Adding these elements would align well with NeurIPS guidelines and contribute to responsible research practices in the field.

---

> ### Author Rebuttal · Authors · 2024-08-07
>
> Thanks so much for your detailed feedback! We will now provide a point-by-point response to your questions.
>
> 1. “Scaling to larger datasets would be beneficial.”
>
> **Reply:**  Thanks for your suggestion. Please see that we have added the ImageNet100 and SVHN evaluation results to Table R1 in the general response.
>
> 2. “Can you provide a detailed analysis of the computational overhead [xxx]? memory usage [...]?”
>
> **Reply:**  Our summary of the computational complexity is in Sec. 3.2, Line 188, and further analyzed in Appendix C.1. The backward pass is not significantly impacted by the number of iterations (Fig. A3); 10 iterations of GCA-INCE require only 5% more FLOPS in general response (#2). GCA-UOT even reduces FLOPS by 30% compared to INCE. Regarding memory usage, GCA-INCE (99.82 MB) and GCA-UOT (51.28 MB) are similar to INCE (51.28 MB) using the same ResNet18 model.
>
> 3. “Hyperparameter sensitivity”
>
> **Reply:** Thank you. In the main text, we summarize the ablation study of the sensitivity of α and β in Sec. 6.2. Additionally, we provide the ablation study of epsilon in Fig. U2. We show the impact of the iteration number and provide the discussion for chosen λ and q in GCA-RINCE in Fig. U3. We hope our efforts alleviate your concern about the hyperparameters sensitivity.
>
> 4. “In the domain generalization [...], how sensitive is the performance to the choice of α and β in the customized transport plan?”
>
> **Reply:**   As shown in Fig. 2, increasing the difference between α-β enhances classification accuracy, where α is the weight for same-domain samples and β for different domains. This is likely due to domain-informative constraints guiding the model during pretraining. Larger weights on same-domain samples benefit domain generalization.
>
> 5. “Did you try comparing GCA with other alignment-based [...] methods? If so, what were the results? How does GCA differentiate itself [...]?”
>
> **Reply:**   Thanks. In Wang, et al, they consider a different setup where they use OT to align seen and unseen images to achieve zero-shot performance. In Balaji et al, they introduce a Wasserstein metric to perform adversarial learning and domain adaptation. It’s hard to compare their work with us because they are not following the same  contrastive learning framework. However, their methods reveal additional settings where contrastive learning can be combined with different adversarial or generation tasks.
>
> 6. “ Can you elaborate on any potential limitations or failure cases of the GCA approach?”
>
> **Reply:** For large datasets, GCA methods may consume more resources due to numerous alignment plans. Single-step methods might perform better and converge faster. Additionally, incorrect hyperparameters like epsilon can cause GCA to fail.
>
> 7. “Ablation on the number of alignment steps.”
>
> **Reply:**  We use a convergence criterion for stopping the alignment. However, we found that typically 5 steps are sufficient for convergence.  Based upon your suggestions, we conducted an ablation study on CIFAR-10 (Fig. U2) showed that accuracy and cluster compactness stabilize after 5 iterations.
>
> 8. “The empirical evaluation of unbalanced OT  is limited.  [...]. More extensive experiments or analysis demonstrating the specific benefits of unbalanced OT over balanced OT in different scenarios”
>
> **Reply:**   Thanks. We ran additional long-tail classification experiments to show the performance of unbalanced OT (Table R2 in general response). These experiments show that GCA-UOT outperforms other baselines and highlight another application where unbalanced OT outperforms other balanced alignment approaches.
>
> 9.  “ how should practitioners choose between different variants [...]? Can you provide more concrete guidelines [...]? Are there specific task characteristics [...]?”
>
> **Reply:** Choosing the right method depends on specific constraints. GCA-UOT consistently performs best across tasks, and GCA-RINCE generally outperforms GCA-INCE, especially with noisy data. With fewer classes and good augmentations, all methods perform comparably (see CIFAR-10 in Table 1). However, for noisy or unbalanced views, GCA-UOT offers the flexibility to add accommodating constraints.
>
> 10.  “Is there a point of diminishing returns [...]”
>
> **Reply:** Yes, increasing iterations in our multistep objective shows no performance improvement beyond a certain point (Fig. U2). On CIFAR-10, this plateau occurs early, typically around 5 iterations.
>
> 11. “Have you explored using different transport plans [...]?”
>
> **Reply:**  In our experiments, we use a single transport plan based on matching constraints: a diagonal target for most experiments and a block constraint for domain adaptation. Using multiple matching constraints for one dataset is an interesting idea!
>
>
> 12. “Can you provide more details on how GCA improves robustness [...]?”
>
> **Reply:** Thanks for the questions. Fig. U3 shows that tuning the hyperparameter q close to 1 gives GCA-RINCE symmetry properties, making it more robust to strong augmentation. Additionally, setting the divergence in GCA as symmetry loss (GCA-RINCE) , we see through Lemma 7 (Appendix A.7) that we can enhance robustness to noisy augmentations. This is supported both theoretically and empirically.
>
> 13.  “Are there specific types of noise [...]”
>
> **Reply:** Thanks. Our results in Tables A2-A4 (Appendix B.2) show that GCA significantly improves under strong crop and large erase conditions, indicating its effectiveness in mask distribution recovery.
>
> 14.  “The authors have not explicitly addressed limitations [...]”
>
> **Reply:**  Thanks. We have updated our discussion of limitations by incorporating your points: 1) extra computational overhead potentially increasing energy consumption, and 2) potential misuse in privacy-sensitive applications.

---

> > ### Author Response · Authors · 2024-08-13
> >
> > Dear Reviewer ih3d,
> >
> > Thank you for your initial feedback on our work. We wanted to follow up and see if there were any additional questions or points of clarification that we could address. Your insights are highly valuable to us, and we are eager to make any necessary improvements based on your suggestions.

---

### Official Review · Reviewer_96Mt · 2024-07-14

**Soundness:** 2
**Presentation:** 2
**Contribution:** 1
**Rating:** 4
**Confidence:** 4

**Summary:**

This paper proposed to view contrastive learning (CL), a popular framework for learning data representation in machine learning. Specifically, the work builds on some recent works that view CL as an alignment problem with optimal transport, showing some previous popular CL frameworks are the special form of this new framework, named generalized contrastive alignment (GCA). Leveraging the theory of unbalanced optimal transport, the authors also introduced an unbalanced CL loss to handle outliers. Empirical benchmarks on the CIFAR-10 and CIFAR-100 datasets are performed to show the effectiveness of GCA.

**Strengths:**

1. The methodology and theory introduced is sound and based on the existing theory of optimal transport and its entropic regularization version.

2. The writing is good and easy to follow. The authors also seem to have done a thorough literature review related to their work.

**Weaknesses:**

1. **Major: there is a large overlap, not in the writing, but in the idea and content of this work and Shi et al (2023).** In particular, the most important one that I want to point out is Algorithm 1 in this work and Algorithm 1 in Shi et al (2023). If one replaces the Bregman div $d_\Gamma$ and $d_M$ by KL divergence into the proximal loss of the inner optimization (eq 8), and $h(P)$ as in the indicator function, one recovers exactly the proposed loss in Shi et al (2023), Eq 8. In the empirical evaluation part, I believe the authors of this work used the aforementioned setting as well, hence I wonder what is the major difference between the main proposed method and that of Shi et al (2023). The idea of using unbalanced OT and its connection to MoCo also has been mentioned in Shi et al (2023, end of Section 3.2).

2. The empirical evaluation is interesting, but I do not find it enough. I believe the authors should have included Shi et al. (2023) as a baseline. The evaluation also should have been done equally on larger dataset such as ImageNet.

3. This is more philosophical point: I am not sure whether if we want the transport plan $P$ to be smooth. Therefore I wonder how did the authors tune their entropic regularization $\varepsilon$ parameter? In computational optimal transport there is a well-known tradeoff between the sparsity of the solved transport plan (in the proximal step of the inner problem) and the stability of scaling operation: if the value $\varepsilon$ (which makes the plan more sparse) the operation will be quite unstable. The Sinkhorn update will run better and faster with larger $\varepsilon$, but the transport is not sparse anymore. I also suggest the authors do a visualization of the quality of solution $P$, similar to what has been done in Figure 3 of Shi et al (2023).

- Small nitpick: Eq (1) missing expectation wrt to samples.


Liangliang Shi, Gu Zhang, Haoyu Zhen, Jintao Fan, and Junchi Yan. Understanding and generalizing contrastive learning from the inverse optimal transport perspective. In International Conference on Machine Learning, pages 31408–31421. PMLR, 2023.

**Questions:**

See weaknesses.

**Limitations:**

Limitation is discussed, but an important point I raised on the weaknesses section is missing.

---

> ### Author Rebuttal · Authors · 2024-08-07
>
> We sincerely thank you for your comments and detailed feedback. We appreciate the opportunity to discuss the contributions of our work in connection to Shi et al. and also expand our comparisons based upon your suggestion.
>
> 1. Discussion of overlap with Shi et al.
>
> **Reply:**  Thanks for your comment and the opportunity to discuss the differences between our work and theirs. As you point out, Shi et al. have provided initial connections between INCE and optimal transport to build a multistep algorithm for alignment using Sinkhorn iterations. In our work, in order to incorporate more general losses and alignment objectives, we have made a number of novel contributions:
>
> - **New algorithm for generalized alignment for contrastive learning:** To allow for more generalized losses, our algorithm allows the intersection of new constraint sets to be iteratively solved, while previous work mainly focuses on the solution of alignment exclusively through Sinkhorn iterations.
>
> - **Novel approach for unbalanced OT-based alignment:** We leverage a rich body of work in OT to introduce a variant of GCA that relaxes the constraints on the distribution penalty (Sec. 3.3). By converting the hard penalty (constraint sets) into the soft regularization terms, our GCA-UOT method achieves high classification accuracy (Table 2) and faster convergence than INCE (Fig. A3), linking OT literature with optimization and contrastive learning.
>
> - **Connections and a multistep variant of RINCE:**  By building a more generalized form of alignment, we demonstrate that it's possible to develop connections to the Robust INCE loss, RINCE. This equivalence enables us to develop a multistep RINCE variant that performs better with corrupted views.
>
> - **Novel results in domain generalization through block-diagonal matching constraints:** By changing the target plan to have block diagonal structure, we can absorb the domain information (Sec. 6). Adding domain-specific matching constraints can improve the pre-training model and enhance classification accuracy in cross-domain generalization tasks.
>
> - **New theory and insights:**  We provide the illustration and prove for the convergence of our more generalized algorithms, not just for the KL divergence in sinkhorn situations, but for other Bregman divergences, which is not shown in previous algorithms and develop new results to explain why running GCA could lead to better uniformity and alignment.
>
> In summary, our GCA framework provides a foundation for addressing a wider range of potential issues in contrastive learning.
>
> 2. Difference between GCA-UOT and the idea of “unbalanced matching” in Shi et al.
>
> **Reply:**   Thanks for your question. The usage is actually quite different. In our case, we are using the term “unbalanced OT” to refer to a large body of work [1,2] in the OT literature that focuses on relaxing the constraint on the distribution matching across the source and the target domain. This often involves building an unconstrained optimization objective and relaxing our constraints on distribution matching. We find that this relaxation has two main advantages: (i) improved accuracy  and (ii) improved complexity and faster speed of convergence (Fig. A3).
>
> In contrast to our method, Shi et al. introduced the idea of “unbalanced matching” which models the fact that the two different encoders in twin network approaches like Moco consider views from two different encoders (often tied via a momentum term).  Our approach, GCA-UOT, employs unbalanced optimal transport which aims to convert hard penalty (constraint sets) into the soft penalty like L2 regularization terms.
>
> [1] Xu, M., & Gould, S. (2024). Temporally Consistent Unbalanced Optimal Transport for Unsupervised Action Segmentation. arXiv:2404.01518.
>
> [2] De Plaen, H., et al. (2023). Unbalanced optimal transport: A unified framework for object detection. CVPR, 3198-3207.
>
> 3. “I believe the authors should have included Shi et al. (2023) as a baseline.”
>
> **Reply:**  Thanks. Indeed, there are some differences in the implementation of Shi et al. version of MS-INCE in comparison to ours as we use a dual form of the OT objective and Shi uses the Sinkhorn algorithm. We tried to obtain their exact implementation but couldn’t find code online so we implemented it and ran this method as another baseline. Please see the (see Table R1, general response) below for the comparison with their IOT method on ImageNet100, SVHN. We are working on adding this baseline for all of the results in the paper.
>
> 4. “Evaluation also should have been done equally on a larger dataset such as ImageNet.”
>
> **Reply:**  Thanks for your suggestion. Rather than focusing on larger datasets with more classes like ImageNet, we chose to explore new settings like domain generalization and robust variants of CL where we could demonstrate the versatility of the approach. Based upon the reviewer's suggestions, we ran additional experiments on ImageNet-100 and SVHN (see Table R1 in the general response). Our results show that our alignment methods improve over standard contrastive losses for the same backbone. Consistently, we find that UOT gives even further improvements over the other methods.
>
> 5. “How did authors tune their entropic regularization ε  parameter? [...] I also suggest the authors do a visualization of the quality of solution P, similar to what has been done in Figure 3 of Shi et al”
>
> **Reply:**  Thank you. We have computed the same plots (Fig. U1) and include an analysis of how the epsilon changes performance.  A smaller epsilon results in a sparser optimal transport plan but potentially leads to numerical instability. We chose a small epsilon (ε=0.2) for CIFAR-10 through cross-validation, however, we find that our alignment method is not very sensitive to this choice. We also examined different ε values for our unbalanced OT method on CIFAR-10 (Fig. U2). The results suggest that an ε range of 0.2-0.6 achieves good compactness and high accuracy.

---

> > ### Comment · Reviewer_96Mt · 2024-08-12
> >
> > Thank you for your rebuttal. I increased my evaluation to 4 to reflect the newly added materials; the evaluation still leaning Reject for the following reason:
> >
> > 1. The result in Table R1 is actually quite mixed: one could see that given enough pretraining iterations, the performance of all 4 methods are very close (SHVN on 500 epoch, third column -- note I have no idea why the authors skip the CIFAR100 dataset). For ImageNet I can understand it takes quite a bit of time to train the network, but I suspect if taking ResNet50 at 500 epochs results would be the same. I also do not understand why IOT of Shi et al. (2023) could perform so badly compared to other 3 on ImageNet.
> >
> > 2. I still hold my opinion that the novelty here is somewhat limited, in the sense that while extending the framework of Shi et al. (2023) is non trivial, it is as hard as extending from classical OT to unbalanced OT setting.
> >
> > I also observed other reviewers have some interesting questions and comments, especially Reviewer TBTc. I therefore look forward to see the answers to those comments as well.

---

> > > ### Author Response · Authors · 2024-08-13
> > >
> > > Dear Reviewer 96Mt,
> > >
> > > Thank you for your continued engagement with our work and for revisiting your evaluation based on our rebuttal. We appreciate the constructive feedback and would like to address the specific concerns you have raised.
> > >
> > > 1a. “ given enough pretraining iterations, the performance of all 4 methods are very close (SHVN on 500 epoch, third column -- note I have no idea why the authors skip the CIFAR100 dataset). For ImageNet I can understand it takes quite a bit of time to train the network, but I suspect if taking ResNet50 at 500 epochs results would be the same.
> > >
> > > While it is true that the INCE-based methods provide similar results, we find that our unbalanced OT approach, GCA-UOT, consistently provides improvements over these other methods. We note that this is out-of-the-box without tuning of the hyperparameters for the alignment method. Thus, this demonstrates that our approach generalizes well and can be used with little to no tuning in new datasets.
> > >
> > > Please note that we did not add CIFAR100 to the response because we provide these results in Table 2 in the main text.
> > >
> > > 1b.  “I also do not understand why IOT of Shi et al. (2023) could perform so badly compared to other 3 on ImageNet.”
> > >
> > > We applied both our method and the Shi method out of the box, using the same parameters as described in their paper, without tuning further. Perhaps with some small tuning their implementation of MS-INCE and our dual form might be more comparable.
> > >
> > > 2. Novelty of the Contribution
> > >
> > > By providing a more general framework for contrastive alignment, our work allows for numerous extensions, including UOT and robust INCE-based losses, as well as block wise constraints in domain generalization. We think this general formulation is key for unlocking additional applications in the future as well. Shi’s work has only considered Sinkhorn-based INCE alignment and thus we see the possibilities with our framework as providing a major advance over the existing work.

---

> > > > ### Comment · Reviewer_96Mt · 2024-08-14
> > > >
> > > > > Please note that we did not add CIFAR100 to the response because we provide these results in Table 2 in the main text.
> > > >
> > > > Just a comment: Table R1 in the rebuttal is the result for Resnet-50; Table 2 is the result for Resnet-18, and there is no IOT of Shi et al. 2023 for comparison in that table in the main text. There is a strong discrepancy in prediction accuracy between a pretrained Resnet-18 and Resnet-50 -- this is now a well-known fact in the deep learning community.

---

> > > > > ### Author Response · Authors · 2024-08-14
> > > > >
> > > > > Thank you for your comment. We will include the CIFAR100 experiments and comparisons with IOT of Shi et al. (2023) in the main text in our revised submission.

---

### Author Rebuttal · Authors · 2024-08-07

We thank the reviewers for their feedback. We appreciate that the reviewers thought that our work “provides an interesting new perspective on what SSL does” (Reviewer **TBTc**) and “the methodology and theory introduced is sound”  (Reviewer **96Mt**), provides “ a solid theoretical foundation”  (Reviewer **VXuY**). Additionally, the reviewers remark that approaches “flexibility [...] opens up new possibilities for tailoring representations to specific tasks or domains.” (Reviewer **ih3d**).

In our general response, we would like to: (1) highlight the key differences between our approach and prior work [reviewers **96Mt, TBTc**]; (2) address the concern of computational complexity [reviewers **TBTc, ih3d, VXuY**]; (3) provide additional larger-scale experiments [reviewers **96Mt, VXuY**].

**1. Clarify contributions from related work [96Mt, TBTc]:**

Previous work from Shi et al. provides connections between InfoNCE and optimal transport using the Sinkhorn method. In contrast to this work, we make a number of novel contributions:

-  **New algorithm for generalized alignment for contrastive learning:**  Our algorithm handles more generalized losses by iteratively solving new constraint set intersections, whereas previous work focused solely on alignment via Sinkhorn iterations.

- **Novel approach for unbalanced OT-based alignment:**  We leverage a rich body of work in OT to introduce a variant of GCA that relaxes the constraints on the distribution penalty (Sec. 3.3). By converting the hard penalty (constraint sets) into the soft regularization terms, our GCA-UOT method achieves high classification accuracy (Table 2) and faster convergence than INCE (Fig. A3), linking OT literature with optimization and contrastive learning.

- **Connections and a multistep variant of RINCE:**  By building a more generalized form of alignment, we demonstrate that it's possible to develop connections to the Robust INCE loss, RINCE. This equivalence enables us to develop a multistep RINCE variant that performs better with corrupted views.

- **Novel results in domain generalization through block-diagonal matching constraints:**  By changing the target plan to have block diagonal structure, we can absorb the domain information (Sec. 6). Adding domain-specific matching constraints can improve the pre-training model and enhance classification accuracy in cross-domain generalization tasks.

- **New theory and insights**:   We prove the convergence of our more generalized algorithms, not just for the KL divergence in sinkhorn situations, but for other Bregman divergences, which is not shown in previous algorithms. And we develop new theorems to explain why running GCA could lead to better uniformity and alignment.

In summary, our GCA framework provides a foundation for addressing a wider range of potential issues in contrastive learning, by demonstrating how to create a variety of different contrastive methods and bringing advanced OT approaches to bear in representation learning.

**2. Computational complexity [TBTc, VXuY, ih3d]:**

About the additional complexity of our method, we provide analysis and empirical evidence to show the complexity (Sec. C1) and compare the running time of different methods (Fig. A3) in the appendix. Our results show that GCA iterations only slightly increase the computational complexity, while GCA-UOT is faster than INCE due to the improved symmetry and smoothness of the loss. To add to this, we record the floating point operations per second (Flops) of running GCA methods. We find that GCA-INCE (6.65 MFlops) has 5% more Flops than INCE (6.31 MFlops), while GCA-UOT saves 30% Flops (4.54 MFlops).  These results prove that our GCA-UOT method is not only superior in terms of accuracy but also in speed.

**3. Additional Experiments**

- **Experiments on larger datasets:**  Following the reviewer's suggestions, we ran additional experiments on ImageNet-100 (Table R1) and SVHN (Table R1), and compared with both INCE and Shi et al. (IOT) methods as a baseline. There is no code available from Shi et al. so we implemented their method. We found that our GCA-UOT achieves the highest accuracy on both datasets.

- **Sensitivity analysis:**  Additionally, we provide ablation studies to hyperparameters including the epsilon effect on transport plan (Fig.U1) **[96Mt]**, the iteration number (Fig. U2) **[ih3d, VXuY]**, the effect of q and λ on strong noisy augmentations Fig. U3 **[VXuY]**.

- **Additional examples of UOT:**  To provide more scenarios of GCA-UOT over balanced OT **[VXuY]**, we test our method on the problem of longtail image recognition with CIFAR100-LT (Table R2).

We hope our additional experiments would address your concerns. Please let us know if there are additional results that you would like to see and we will do our best to provide them.

**Results**

**Table R1. Comparison of GCA methods with other baselines on SVHN and ImageNet100 datasets.** For SVHN, we pre-trained ResNet50 models using the Adam optimizer with a learning rate of 3e-4 for both 200 and 500 epochs. For ImageNet100, we pre-trained a ResNet50 model for 100 epochs using the same optimizer and learning rate. The classification accuracy was then evaluated using a linear layer readout trained 100 epochs.

|  | **SVHN**  |    | **ImageNet100**  |
| -- | -- | -- | -- |
|**Epochs**| **200**|**500**| **100**|
|**UOT**| **89.98**|**91.85**|**68.63** |
|**MSINCE**|86.00|89.86|67.60|
|**INCE**|86.60|89.79|67.93|
|**IOT(Shi​​​​)**|85.19|90.01|66.07|

**Table R2. Comparison of GCA methods with other baselines on longtail recognition (CIFAR100-LT).** We pre-trained models for 500 epochs using the Adam optimizer with an imbalance factor of 0.1 on the CIFAR100 long-tail dataset. The classification accuracy was then evaluated using a linear layer readout trained 200 epochs.
|**Method**|**CIFAR100LT**|
| -- | -- |
|**UOT**|**35.96**|
|**MSINCE**|33.20|
|**INCE**|33.72|
|**ShiINCE**|32.55|

---

### Decision · Program_Chairs · 2024-09-25

**Decision:**

Accept (poster)

**Comment:**

This paper focuses on the theoretical foundations of contrastive learning (CL) for learning data representations. The authors propose a framework called Generalized Contrastive Alignment (GCA), which connects the noise contrastive estimation loss to distribution alignment using entropic optimal transport. Building on this, the authors introduce variants of self-supervised learning losses to handle outliers. The empirical evaluation on CIFAR-10, CIFAR-100, ImageNet-100, and SVHN (added during the rebuttal) demonstrates the effectiveness of the proposed method.

The reviewers raise concerns about the overlap with existing work and the absence of a discussion on related work (Reviewers 96Mt, TBTc), the structure of the paper (Reviewer TBTc), computational complexity (Reviewers TBTc, VXuY, ih3d), and the practical applicability of the proposed algorithm (Reviewer VXuY), among others. During the rebuttal, these concerns were thoroughly discussed, and the authors effectively addressed many of them. Given the overall positive reception of the proposed framework, we recommend accepting the paper. However, if accepted, the authors should include the requested revisions from the reviewers, such as the discussion of overlap with related work and improvements to the paper's structure.